# Experimental Static and Dynamic Characteristics of Recycled Waste Tire Rubber Particle–Cement–Sand Composite Soil

**DOI:** 10.3390/ma15248938

**Published:** 2022-12-14

**Authors:** Zhaoyan Li, Liping Zhang, Haiyang Zhuang, Qi Wu

**Affiliations:** 1Institute of Engineering Mechanics, China Earthquake Administration, Harbin 150080, China; 2Institute of Geotechnical Engineering, Nanjing Tech University, Nanjing 210037, China

**Keywords:** cement–rubber–sand composite soil, shear strength, dynamic shear modulus, damping ratio, indoor geotechnical test

## Abstract

To investigate the static and dynamic characteristics of rubber–sand composite soil (RS soil) reinforced with cement, a series of triaxial compression tests and resonant column tests was performed by considering the influence of rubber content (10%, 20%, 30%, 40%, and 50%), cement content (0, 1.5, 2.5, 3.5 and 4.0 g/100 mL), and effective consolidation confining pressure (50, 100, and 150 kPa). Compared with the RS soil, the addition of cement significantly improved the shear strength of a cement–rubber–sand composite soil (RCS soil), based on an undrained shear test. The increase in cement content not only makes the elastic modulus and cohesion of the RCS soil increase but also reduces the internal friction angle of the RCS soil. With the increase in rubber content, the failure of the RCS soil samples changes from strain-softening to hardening, and the prediction equation of the initial elastic modulus of the RCS soil is given herein when the recommended cement content is 3.5 g/100 mL. The effects of rubber content, cement content, and effective confining pressure on the dynamic shear modulus and damping ratio of the RCS soil were studied via the resonant column test. The test results show that the increase in rubber content slows down the modulus attenuation of the RCS soil, but increases its damping ratio. The test results also show that the increase in cement content makes the bonding force between particles greater so that the modulus attenuation of the RCS soil becomes slower and the damping ratio is reduced. At the same time, according to the change rule of the maximum dynamic shear modulus of the RCS soil with the rubber content, when the recommended cement content is 3.5 g/100 mL, an empirical formula and recommended value of the shear modulus *G*_max_ of the RCS soil are proposed.

## 1. Introduction

With the increasing number of private cars and the rapid development of logistics and the freight industry, the production of waste rubber tires increased year by year. The improper disposal of waste tires is likely to cause environmental pollution and the waste of resources. The rubber particles created from waste tires have the advantages of light density, strong compressibility, and good durability. Therefore, waste rubber tire particles can be mixed with different materials to improve the performance of raw materials. Using a mixture of rubber and concrete can significantly improve the durability of concrete [1,2,3]. The mixture of rubber and sand can significantly improve the mechanical properties of soil and can be used as a damping energsy-dissipation material in the field of base isolation, as is the case with the rubber–sand composite soil (RS soil) used in this study. However, it is necessary to overcome its shortcomings while maintaining its light weight, super-elasticity, and high damping characteristics. The improved RS soil can be used for a foundation treatment of slopes, retaining walls, and roadbed engineering, and can also be used as an isolation layer.

Through a large number of triaxial tests, direct shear tests, and hollow torsional shear tests, scholars have studied the influence of a rubber addition on the shear strength of rubber–sand mixtures, and there are different views on the influence of rubber content. Rao et al. [4] think that the incorporation of rubber can improve the shear strength of rubber–sand mixtures. The test results reported by Sheikh et al. [5], Noorzad et al. [6], and Liu Qifei [7] showed that the shear strength of rubber–sand mixtures decreased significantly with the increase in rubber content. The main reason for the above differences is that the rubber shapes used by the researchers are different. In general, the addition of a strip or fibrous rubber can improve the strength of soil to a certain extent, while the addition of granular rubber will reduce the shear strength of rubber–sand mixtures. Therefore, Zornberg et al. [8] and Li et al. [9] considered that the mechanical properties of rubber–sand mixtures could be improved by optimizing the aspect ratio of the rubber debris. In addition to rubber content, scholars also explored the influence of other factors. For example, Li et al. [10] studied the difference in the rubber–sand interface state caused by different rubber particle types; Liu Fangcheng’s research group [11,12,13,14,15] explored the influence of the initial state, confining pressure, rubber content, relative density, and particle size ratio on the internal friction angle, cohesion, initial tangent modulus, and Poisson’s ratio of rubber–sand mixtures through a variety of test methods. From research on the development of the stress and strain of rubber–sand mixtures by Deng ’ an et al. [16], Ehsani et al. [17], and Zhang Tao et al. [18], the results show that the hardening level of the stress–strain curve increases with the increase in confining pressure, while the failure mode shows a trend of transition from “brittleness” to “toughness” with the increase in rubber content.

Research into the dynamic characteristics of rubber–sand mixtures requires different test methods according to the strain range that is studied. In the small-strain range (from 10^−6^ to 10^−4^), a resonant column or bending element [19,20] is usually used for testing, while in the large-strain range (from 10^−3^ to 10^−2^), a cyclic triaxial apparatus or hollow torsion shear apparatus [21,22] is used for testing. At present, scholars [23,24] have drawn consistent conclusions on the influence of the main dynamic parameters of rubber–sand mixtures in the small-strain range. The first conclusion is that the increase in cement content reduces the initial dynamic shear modulus of rubber–sand mixtures but increases the initial damping ratio. The second conclusion is that the increase in confining pressure increases the maximum dynamic shear modulus of rubber–sand mixtures but reduces the initial damping ratio. In addition, the research groups of Senetakis [25,26,27,28] studied the influence of various factors on the dynamic shear modulus and damping ratio of rubber–sand mixtures through a resonant column test and put forward an empirical formula of dynamic shear modulus and the damping ratio of rubber–sand mixtures. These factors include the saturated or dry state of the sample, sand particle size characteristics, consolidation time, rubber particle content, strain amplitude, and sample preparation methods. Therefore, the research results published by Senetakis’s groups have a high reference value for the study of the dynamic characteristics of rubber–sand mixtures in the small-strain range. Liu Fangcheng’s research group [29,30,31] used a variety of research methods to study the effects of many factors on the dynamic characteristic parameters of rubber–sand mixtures; they specifically pointed out that there is a coupling effect between the rubber–sand particle diameter ratio, water content, and rubber content on the dynamic shear modulus and damping ratio. Feng et al. [32] pointed out that the maximum dynamic shear modulus of rubber–sand mixtures can be predicted using the modified Hardin formula, and the attenuation law of the dynamic shear modulus ratio with shear strain amplitude can be fitted using the Martin–Davidenkov model [33]. Li Xiaoxue et al. [34] reported that the growth law of the damping ratio with shear strain amplitude can be fitted by the empirical formula proposed by Chen Guoxing.

Studies have shown that the addition of rubber particles to sand will improve the shear strength characteristics of sand to a certain extent, reduce its elastic modulus, and change its failure mode [5,6,7]. However, studies have also shown that a rubber particle–soil mixture is still an isotropic material; it has a small shear modulus in the horizontal direction, and the vertical compression modulus is also small, which leads to large compression deformation under static load. Therefore, its elastic modulus is low, which easily causes instability in the foundations of the structure and increases the frequency of the left and right swaying of the structure under seismic load [35]. In order to solve the above problems, this paper proposes to add cement as a cementing and curing agent to the RS soil to improve the shear strength. On this basis, the effects of different rubber contents, cement contents, and confining pressure on the internal friction angle *c* and cohesion *φ* of RCS soil were studied with an undrained shear test, and the change of its failure mode under different working conditions was analyzed. At the same time, the variation law of the initial elastic modulus *E_0_* of RCS soil according to rubber content and its prediction equation under different test conditions were given. Finally, through a series of resonant column tests, the effects of the RCS soil on the decline curve of dynamic shear modulus, *G*, and the growth curve of the damping ratio, *λ*, under different rubber contents, cement contents, and confining pressures were studied, and the prediction formula of the maximum dynamic shear modulus, *G*_max_, was given.

## 2. Test Scheme

### 2.1. Test Materials and Instruments

The sand used in this paper is ISO-standard sand from Fujian, China. This sand is widely used in various geotechnical tests. Its particle sizes range from 0.075 mm to 2 mm, and its specific density, *G*_s_, was 2.71. The waste tire rubber particles used in the test were purchased from merchants specializing in the decomposition of waste tires. The particle size ranged from 2 mm to 3 mm, and the specific gravity, *G*_s_, was 1.73. The cement used in the test is Nanjing Yuhua cement, which is ordinary 425 Portland cement. The schematic diagrams and gradation curves of standard sand particles and crumb rubber particles after the crushing of waste tires are shown in Figure 1 and Figure 2. The specific values of particle gradation are listed in Table 1.

The instruments used in this test are the DYNTTS GDS triaxial test instrument (Figure 3a) and TSH-100 high-precision fixed free resonant column test system (Figure 3b) in the Institute of Geotechnical Engineering, Nanjing Tech University, as shown in Figure 3. Axial strain is the ratio of the axial displacement, produced during the triaxial compression test, to the initial height of the specimen before the triaxial compression test. During the resonant column test, the bottom of the sample is fixed to the instrument base, and the top of the sample is connected to the suspension torsion drive system; therefore, it can be considered that a lumped mass is added to the top of the sample and, through this mass block, torsional harmonics with constant amplitude can be applied to the sample within a certain frequency range [36].

### 2.2. Sample Preparation and Test Scheme

The test sample is a cylindrical sample with a diameter of 50 mm and a height of 100 mm [37,38,39]. The relative density of the sample is 70%. Firstly, the waste tire rubber particles and the dried standard sand particles were mixed evenly, according to the mass ratio (10%, 20%, 30%, 40%, 50%); then, the quantitative cement value (from 0 g/100 mL to 4.0 g/100 mL) was added according to the volume ratio. Finally, an appropriate amount of water was added, and the mixture was mixed evenly according to a water–cement ratio of 1:3 [40]. The prepared RCS soil sample was placed in the curing box for two days after curing, and testing was carried out after 7 days of curing. The specific experimental conditions are shown in Table 2.

According to previous studies, the most difficult part of the preparation of the rubber and sand mixture is to ensure the uniformity of rubber particles in the sand particles, which is key to ensuring the reliability of the test results. In this experiment, due to the addition of a small amount of cement as a binder, small cement particles will adhere to the surface of rough rubber particles. After adding water and stirring, the cement, rubber, and sand are bonded to each other; this means that the rubber particles of the mixture are uniformly distributed, making the sample preparation method simpler and more convenient. The prepared specimens and their cross-sections are shown in Figure 4.

## 3. Analysis of Triaxial Compression Test Results

### 3.1. Failure Mode Analysis of Cement–Rubber–Sand Mixtures

In the triaxial compression tests, RCS soil samples with different rubber contents showed different mechanical properties and failure modes.

Excess pore water pressure, *u*, may occur in consolidated undrained shear tests [41] The pore pressure coefficient is *A* = *u*/(*σ*_1_ − *σ*_3_); the value of *A* can reflect the dilatancy (shrinkage) characteristics of soil failure. The pore pressure coefficient, *A*, is constantly changing in the undrained shear test: elastic body, *A* = 1/3; dilatancy of the soil, *A* < 1/3; shear shrinkage of the soil, *A* > 1/3 [41]. The variation curve of the pore pressure coefficient, *A*, with the axial strain of two different rubber contents is shown in Figure 5. It can be seen from Figure 5 that when the rubber content *X* < 20%, the pore pressure coefficient *A* of the cement–rubber–sand mixed soil is less than 1/3, indicating that the RCS soil has only a dilatancy effect. When *X* ≥ 20%, with the increase in axial strain, the pore pressure coefficient of RCS soil first indicates shear contraction (*A* > 1/3) and then shear dilatation (*A* < 1/3). Figure 6 shows the stress state and failure mode of RCS soil under different working conditions [37]. The stress state of the specimen during the test is shown in Figure 6a. It can be assumed that when the rubber content *X* < 20%, the force transmission skeleton in the sample is mainly from sand–sand contact. The curing effect of cement further enhances the interlocking effect between sand particles, which leads to improvement in the overall strength of the sample. The failure mode is close to that of pure sand, and the strength decreases rapidly after the formation of a 45° shear failure surface, showing strain-softening failure; that is, there is an obvious peak point in the stress–strain curve. The failure mode of strain softening is shown in Figure 6b. When the rubber content *X* ≥ 20%, the force transmission mode of the sample skeleton becomes primarily controlled by the large deformation of the rubber particles in the sample; the failure mode of the specimen was gradually transformed into lateral bulging deformation failure. The failure pattern is shown in Figure 6c,d and the whole sample exhibits a shear contraction trend, showing strain-hardening failure; that is, there is no obvious peak point in the stress–strain curve.

### 3.2. Stress–Strain Relationship of Cement Rubber Sand

The stress–strain curve of the RCS soil with different rubber contents is shown in Figure 7. It can be seen from the figure that when the content of rubber particles in the RCS soil is small (*X* < 20%), the stress first increases to the peak value and then decreases with the increase in strain. With the increase in rubber content (*X* ≥ 20%), the stress increases with the increase in strain, and the growth rate decreases gradually. The increase in rubber content also reduces the stress–strain curve of the RCS soil and the shear strength. This is because the particle contact mode in the RCS soil with low rubber content is still sand–sand contact, and the increase in strain will make the particle bite force reach its limit so that the shear band strength of the mixed sample decreases rapidly. With the increase in rubber content, most particles of the RCS soil are in contact with the rubber. Due to the high compressibility of rubber, volume deformation of the sample occurs, meaning that the gap of the sample is reduced and the contact area between sand and rubber becomes larger, resulting in increased friction and bite force between particles, which can inhibit the decrease in strength after the failure of the cemented structure. Due to the high compressibility of rubber particles, when the rubber content is large (*X* ≥ 20%), the RCS soil exhibits a single shear shrinkage failure mode during the shear process.

It can also be seen from Figure 7 that the confining pressure has little effect on the stress–strain curve of the RCS soil sample. This is due to the solidification effect of cement, which means that the particles of the sample are more closely arranged so that the strength of the sample becomes larger. Therefore, the change in confining pressure has no obvious effect on the stress–strain value of the RCS soil.

The stress–strain curve of RCS soil samples with different cement contents is shown in Figure 8. It can be seen from the diagram that with the increase in cement content, the higher the stress–strain curve of the RCS soil, the higher the shear strength of the RCS soil. This shows that the increase in cement content makes the cementation between particles stronger and greatly improves the shear strength of the RCS soil.

According to the deviatoric stress at failure criterion [37], the deviatoric stress at failure under different rubber content, cement content, and confining pressure scenarios is listed in Table 3. Three conclusions can be drawn from Table 3. The first is that with the increase in rubber content, the smaller the stiffness of the sample, the smaller the deviatoric stress at failure. The second is that with the increase in consolidation confining pressure, the sample particles are more closely arranged, which means that the deviatoric stress at failure is greater when the sample is destroyed. The third is that with an increase in cement content, the bonding bite force between particles increases, and the deviatoric stress at failure increases exponentially.

### 3.3. Influencing Factors and the Prediction Method of the Initial Elastic Modulus

According to the assumption of linear elasticity [38,42], the part of the stress–strain curve in the range of axial strain, *ε*_a_ ≤ 1 × 10^−5^, is linearly fitted, and the slope of the fitted line is the initial deformation modulus, *E*_0_. In 1963, Janbu proposed that the initial elastic modulus, *E_0_*, is related to the confining pressure, and gave the following empirical formula [43]:(1)$$E_{0}=K\cdot \mathrm{Pa}{\left(\sigma_{3}/\mathrm{Pa}\right)}^{n}$$
where *K* and *n* are empirical constants and Pa is the standard atmospheric pressure value (101.325 kPa).

Figure 9 shows the relationship between the *E*_0_~*σ*_3_/Pa of RCS soil when the cement content is 3.5 g/100 mL in this test; the fitting parameters *K* and *n* can be obtained from Equation (1). It can be seen that the curve inclination decreases with the increase in rubber content; when the rubber content exceeds 30%, the fitting curves are basically parallel to each other.

Figure 10 is the relationship between the *E*_0_~*σ*_3_/Pa of RCS soil with 20% rubber content in this test. The fitting parameters, *K* and *n*, are obtained using the empirical formula given by Janbu. It can be seen that the curve inclination increases with the increase in cement content.

Figure 11 shows the variation of the fitting parameters, *K* and *n*, with rubber content. It can be seen that the initial modulus coefficient *K* decreases exponentially with the increase in rubber content, and Formula (2) can be obtained by exponential fitting:(2)$$K=2236\cdot e^{-0.07\cdot X}$$
where *X* is the rubber content.

The value of index *n* fluctuates slightly from 0.7 to 0.73. For a fixed cement content, *n* can be taken as an average of 0.72. Substituting Formula (2) and *n* = 0.72 into the empirical formula, Formula (1), the relationship between the initial elastic modulus *E*_0_ of the RCS soil with 3.5 g/100 mL of cement content and the rubber content *X* is given as shown in Formula (3):(3)$$E_{0}=2236\cdot e^{-0.07\cdot X}\cdot \mathrm{Pa}\cdot {\left(\sigma_{3}/\mathrm{Pa}\right)}^{0.72}$$
where *E*_0_ is the initial elastic modulus, *X* is the rubber content, and Pa is the standard atmospheric pressure (101.325 kPa).

Figure 12 shows the variation of fitting parameters *K* and *n* according to cement content. It can be seen that the initial modulus coefficient *K* increases exponentially with the increase in cement content, and Formula (4) can be obtained by exponential fitting:(4)$$K=175.42\cdot e^{0.323\cdot C}$$
where *C* is the cement content.

The relationship between the actual value of the initial elastic modulus, *E*_0_, and its predicted value is plotted in Figure 13. It can be seen that the scatter points are evenly distributed near the 45° line, and the fitting of R^2^ reaches 0.996, indicating that the predicted value is close to the measured value. It can be seen that the empirical formula of the initial elastic modulus, *E*_0_, is reliable.

### 3.4. Shear Index and Strength Analysis

According to the deviatoric stress at the failure of the RCS soil with different rubber contents and cement contents, the total Mohr’s circle and the effective Mohr’s circle under the corresponding rubber contents are drawn on the *τ*~*σ* coordinate diagram, according to the specification [37,38,39]. Figure 14 shows the Mohr’s circle and corresponding shear strength envelopes of RCS soil with different rubber contents and different cement contents.

According to the above Mohr’s circle, the internal friction angle and the cohesion of different rubber contents and cement contents are shown in Table 4. This shows that the shear strength index of the RCS soil is affected by the rubber content and cement content. The relationship between the shear strength index of the RCS soil, as seen in the internal friction angle and cohesion, and the rubber content is shown in Figure 15. It can be seen from the diagram that the internal friction angle of the RCS soil increases with the increase in rubber content, and the effective internal friction angle is always greater than the total internal friction angle. The internal friction angle increases slowly when the rubber content *X* < 30% and increases rapidly when the rubber content *X* ≥ 30%. The cohesion of the RCS soil decreases with the increase in rubber content, and the effective cohesion is always smaller than the total cohesion. When the rubber content *X* < 30%, the cohesion decreases rapidly, and when the rubber content *X* ≥ 30%, the cohesion decreases slowly.

This phenomenon can be analyzed by the way in which each strength component plays a role in the stress–strain development process of soil, as shown in Figure 16 [39]. The shear strength of cohesive soil consists of three parts: the cohesion component, the friction component, and the dilatancy component. The increase in rubber content will reduce the cementation of the cement between particles, thus reducing the dilatancy component and cohesion component, resulting in a decrease in the measured strength of the RCS soil. This makes the radius of Mohr’s circle decrease; the strength envelope moves down, and the cohesion decreases. At the same time, because of the irregular shape of rubber particles, the increase in its content makes the internal friction angle between the particles increase.

The relationship between the shear strength index, such as the internal friction angle and cohesion, and cement content is shown in Figure 17. It can be seen from the figure that as the cement content increases, the internal friction angle decreases, and the effective internal friction angle is greater than the total internal friction angle. The increase in cement content also makes the cohesion increase, and the effective cohesion is less than the total cohesion. This is because the increase of cement content strengthens the cementation structure between the particles inside the RCS soil, thereby increasing the cohesive component of the RCS soil; because the cement content is very small, the proportion of the total volume of the RCS soil is very small, so the increase in the cement content has little effect on the friction component and the dilatancy component, which doubles the measured strength of the RCS soil. The radius of the stress circle becomes larger at the same time, under different confining pressures, which means that the envelope line moves up as a whole and the cohesion increase greatly. The increase in cement content makes the stiffness of the sample larger, and the influence of confining pressure on it decreases, so that the radius of the Mohr’s circle under different confining pressures changes little, and the slope of the envelope curve decreases, resulting in a decrease in the internal friction angle.

## 4. Analysis of the Dynamic Modulus and Damping Ratio Test Results

### 4.1. Analysis of the Influencing Factors of Dynamic Shear Modulus

According to the resonant column test, the results show that the dynamic shear modulus *G* of RCS soil is related to the cement content, rubber content and consolidation confining pressure of the RCS soil. For the RCS soil, the dynamic shear modulus decreases with the increase in shear strain amplitude.

When the cement content is 3.5 g/100 mL, under the same confining consolidation pressure, the relationship between the dynamic shear modulus, *G,* and the rubber content, *X*, of the RCS soil is shown in Figure 18. It can be seen from the diagram that with the increase in shear strain amplitude, *γ*, the dynamic shear modulus, *G*, of the RCS soil with different rubber contents decreases. The dynamic shear modulus, *G*, decays slowly in the range of 1 × 10^−6^ ≤ *γ* ≤ 1 × 10^−5^, and when the shear strain increases to 1 × 10^−5^ ≤ *γ* ≤ 5 × 10^−4^, the dynamic shear modulus, *G*, decays rapidly. The dynamic shear modulus, *G*, of RCS soil decreases with the increase in rubber content, which is because the increase in rubber content leads to a decrease in sample stiffness and shear resistance. When the rubber content *X* ≤ 30%, the attenuation of the dynamic shear modulus, *G*, is larger, and the attenuation of the dynamic shear modulus, *G*, of the RCS soil is obviously slowed down when *X* > 30%. When the rubber content *X* = 50%, the dynamic shear modulus (*G*) decay is flat. This phenomenon is because when the rubber content *X* is small, the skeleton is mainly sand, and cement, as the cementing agent, creates more cementation between the sand and sand particles. Once the cementation between particles is halted, the sand particles will dislocate from each other, and the dynamic shear modulus, *G*, will decay rapidly. With the increase in rubber content, the deformation of the rubber particles makes the contact area of the particles larger, restrains the mutual dislocation friction between the particles, and makes the dynamic shear modulus, *G*, attenuate slowly.

It can also be seen from Figure 18 that the influence of confining pressure on the attenuation of the dynamic shear modulus, *G*, of RCS soil is basically the same. With the increase in consolidation confining pressure, the occlusion between the particles inside the sample is tighter, which makes it difficult to dislocate the particles, resulting in the greater stiffness of the sample. The greater the stiffness of the sample, the stronger the shear resistance; the dynamic shear modulus, *G*, of the RCS soil with the same rubber content also increases.

### 4.2. Influencing Factors and Prediction Methods of the Maximum Dynamic Shear Modulus

It is generally considered that the shear modulus in the range of *γ*_a_ < 10^−6^ is the maximum shear modulus *G*_max_ [44,45]. According to the hyperbolic model of the soil backbone curve in a small strain range, as proposed by Hardin et al. [46].
(5)$$\tau =\frac{\gamma }{a+b\cdot \gamma }$$

Here, 1/*G* = b∙*γ* + a, that is, there is a linear relationship between shear strain, *γ*, and 1/*G*. According to the resonant column test data, the maximum dynamic shear modulus *G*_max_ of RCS soil can be obtained via the above linear fitting of 1/*G* − *γ*:(6)$$G_{\max}=\underset{\gamma \to 0}{\lim}\frac{1}{a+b\cdot \gamma }=\frac{1}{a}$$
where *a* and *b* are the fitting parameters, which are obtained by fitting the test results.

Figure 19 shows the variations in the maximum dynamic shear modulus *G*_max_ for RCS soil with a rubber particle content, *X*, under different confining pressures, when the cement content is 3.5 g/100 mL. It can be seen that the maximum dynamic shear modulus *G*_max_ of the RCS soil increases with the increase in effective consolidation confining pressure. Under different confining pressures, the maximum dynamic shear modulus, *G*_max_, decreases linearly with the increase in rubber content in the RCS soil; the greater the confining pressure, the more the maximum dynamic shear modulus, *G*_max_, decreases with the increase in rubber content.

Figure 20 shows the variation of the maximum dynamic shear modulus, *G*_max_, of the RCS soil, with the cement content under different confining pressures when the rubber content is 20%. It can be seen that the maximum dynamic shear modulus, *G*_max_, of RCS soil increases with the increase in cement content.

In 1970, Seed [47] established an empirical formula for calculating the maximum dynamic shear modulus, *G*_max_, of sand:(7)$$G_{\max}=21.7\cdot K_{\max}\cdot \mathrm{Pa}\cdot {\left(\sigma_{0}^{\prime }/\mathrm{Pa}\right)}^{0.5}$$
where Pa is the standard atmospheric pressure and *σ*′_0_ is the average principal stress. *K*_max_ is related to the relative compactness of sand, where *K*_max_ = 61∙[1 + 0.01∙(*D*_r_ − 75)].

This test material is a new type of recycled filler made from rubber particles, sand, and cement. Due to the following test studies, a previous scholar simplified Equation (7) into Equation (8), to neglect the effect of relative density [48]. The fitting empirical formula of the maximum dynamic shear modulus, *G*_max_, of the RCS soil and the rubber content, *X*, under the same cement content of 3.5 g/100 mL is given as follows:(8)$$G_{\max}=K\cdot \mathrm{Pa}\cdot {\left(\sigma_{0}^{\prime }/\mathrm{Pa}\right)}^{n}$$
where *G*_max_ is the maximum dynamic shear modulus (MPa) and *K* and *n* are the fitting parameters.

Figure 21 shows the relationship between parameters *K* and *n* and the rubber content when the cement content is 3.5 g/100 mL. By studying the parameters *K* and *n* in the *G*_max_ empirical fitting formula, it is clear that when the rubber content is less than 50%, *n* is approximately 0.5, so it can be considered that *n* is 0.5, and the parameter *K* decreases with the increase in rubber content, which is therefore related to the rubber content, *X*. It can be seen from Figure 21a that parameter *K* decreases linearly with the increase in rubber content, *X*. Figure 22 shows the relationship between parameter *K* and *n* and cement content when the rubber particle content is 20%. It is clear that parameter *K* increases linearly with an increase in cement content. The relationship between parameter *K*, rubber content, *X*, and cement content, *C*, is as follows:*K* = 1752 − 43.8∙*X* + 208∙*C*(9)
where *K* is the parameter in the *G*_max_ empirical formula, and *X* is the rubber content (%).

Formula (9) is substituted into Formula (8). The relationship between *G*_max_ and average principal stress, *σ*_0_’, rubber content, *X*, and cement content, *C,* is given as follows:(10)$$G_{\max}=\left(1752-43.8\cdot X+208\cdot C\right)\cdot {\left(\sigma_{0}^{\prime }/\mathrm{Pa}\right)}^{0.5}$$
where *G*_max_ is the maximum dynamic shear modulus (MPa) of the RCS soil, *X* is the rubber content, Pa is standard atmospheric pressure, and *σ*_0_’ is the average principal stress (kPa), with values of 50 kPa, 100 kPa, and 150 kPa.

Figure 23 shows the comparison between the predicted value of the maximum dynamic shear modulus, *G*_max_, based on the rubber content, the average principal stress, and the actual value obtained from the test. It can be seen that the scatter points are evenly distributed near the 45° line and the fitting R^2^ reaches 0.998, indicating that the fitting value of *G*_max_ is close to the measured value. This paper considers that the empirical formula of *G*_max_ is reliable.

### 4.3. Normalized Modulus and Damping Ratio Analysis

The stress–strain relationship of RCS soil is nonlinear and hysteretic. In order to explore the changes in the attenuation of the dynamic shear modulus, *G*, of the RCS soil with a shear strain of *γ*, the dynamic shear modulus, *G*, was normalized to obtain the normalized dynamic shear modulus ratio *G*/*G*_max_, so as to study the change in the modulus ratio, *G*/*G*_max_, and the damping ratio, *λ*, with the shear strain, *γ*, under different rubber contents, cement contents, and consolidation confining pressures.

In order to quantitatively describe the relationship between the dynamic shear modulus ratio, *G*/*G*_max_, and the shear strain amplitude, *γ*, of RCS soil, the Martin–Davidenkov model [33] with three parameters is selected to describe the modulus characteristics of the RCS soil:(11)$$\frac{G}{G_{\max}}=1-{\left[\frac{{\left(\gamma_{a}/\gamma_{0}\right)}^{2B}}{1+{\left(\gamma_{a}/\gamma_{0}\right)}^{2B}}\right]}^{A}$$
where *A*, *B*, and *γ*_0_ are all the test parameters related to soil properties; when *A* = 1 and *B* = 0.5, the model degenerates into the Hardin-Drnevich hyperbolic model. In the H-D hyperbolic model, *γ_0_* is the reference shear strain, and its value is the corresponding shear strain amplitude when *G*/*G*_max_ = 0.5. Since the fitting of the three parameters in the Martin–Davidenkov model has strong randomness, the definition and value of parameter *γ_0_* in the Hardin-Drnevich hyperbolic model are used in this paper. First, the value of the reference shear strain, *γ_0_*, is calculated according to the Hardin-Drnevich model, then parameters *A* and *B* in the Martin–Davidenkov model are fitted. To identify the relationship between the damping ratio of the RCS soil and the shear strain amplitude under different working conditions, the empirical formula proposed by Chen Guoxing [48] is used for the fitting analysis:(12)$$\lambda =\lambda_{\min}+\lambda_{0}\cdot {\left[1-\left(G/G_{\max}\right)\right]}^{\beta }$$
where *λ*_min_ is the basic damping ratio of the soil, which is related to the properties and consolidation state of the soil. *λ*_0_ and *β* are the shape coefficients of the damping ratio curve, which are the fitting parameters related to the properties of soil.

#### 4.3.1. Effect of the Rubber Content

When the cement content is 3.5 g/100 mL, the variation trends of the modulus ratio *G*/*G*_max_ and the damping ratio, *λ*, with shear strain *γ* under different rubber contents are shown in Figure 24. It can be seen from the figure that under the same consolidation confining pressure, the attenuation law of dynamic shear modulus after the normalization of different rubber contents shows obvious differences. The higher the rubber content of the RCS soil, the higher the modulus ratio, namely, the slower the attenuation rate of dynamic shear modulus is. When the rubber content *X* ≤ 30%, the dynamic shear modulus attenuation curve is close, indicating that when the rubber content is small, the increase in rubber content has little effect on the attenuation of the modulus of the RCS soil. When the rubber content *X* > 30%, the rubber content has an obvious influence on the attenuation of the modulus, and the attenuation rate of the modulus ratio is significantly slowed down. There is a critical shear strain, *γ*_cr_, in the attenuation of the dynamic shear modulus. When the shear strain is less than the critical shear strain (10^−6^ < *γ* < *γ_cr)_*, the attenuation of the dynamic shear modulus is relatively slow. When the shear strain is greater than the critical shear strain (*γ* > *γ_cr)_*, the decay rate of the dynamic shear modulus increases demonstrably. At the same time, with the increase in rubber content, the corresponding critical shear strain *γ_cr_* value also increases. The reason for this phenomenon could be that the rubber particles have a good deformation ability. The increase in the content of rubber particles in the RCS soil improves the elastic deformation ability of the whole soil, which can better delay the rapid attenuation of the shear strength of the soil in an earthquake.

The test results also show that the damping ratio of the RCS soil increases with the increase in rubber content. Due to the strong deformation ability of rubber, the increase in rubber content is bound to increase the damping ratio of the RCS soil. When the shear strain amplitude is in the range of 1 × 10^−6^ ≤ *γ* ≤ 1 × 10^−5^ and the rubber content *X* ≤ 30%, the increase in rubber content has little effect on the increase in the damping ratio. When the rubber content *X* > 30%, the increase in rubber content has an obvious effect on the increase in the damping ratio. This is because in the small strain range, when the rubber content *X* ≤ 30%, the cement has a strong cementing and curing effect on the mixture, the stiffness of the sample is greater, and the deformation of the rubber has a weak effect on the energy dissipation and shock absorption of the sample so that the increase in the rubber content is smaller. When the rubber content *X* > 30%, the cementation and curing effect of cement on the mixture is weakened, and a trend of mutual dislocation between particles is generated. At the same time, the characteristics of a small elastic modulus and large energy consumption of rubber make the deformation capacity of the mixture increase; the energy dissipation capacity increases, and the damping ratio increases significantly. There is also a critical shear strain *γ*_a_ for the increase in the damping ratio of mixtures with different rubber contents [49,50]. The greater the rubber content of the mixture, the greater the critical shear strain, γ_a_, corresponding to the damping ratio curve, and the smaller the damping ratio growth rate, which also leads to the finding that the damping ratio of mixtures with different rubber contents will have an intersection when the shear strain *γ* is about 1 × 10^−4^. The reason for this may be that with the increase in shear strain when the rubber content is great, the rubber particles in the mixture are mainly deformed after the force, and the whole sample is greatly affected by the damping of the rubber particles so that the damping ratio rises slowly. When the rubber content is small, the cementation state between the original particles in the sample is destroyed, the rubber particles are deformed by force, and staggered friction between the particles is gradually generated [51,52]. The particles in the sample are rearranged and a certain amount of energy is consumed. In this strain range, the damping ratio will grow more rapidly.

#### 4.3.2. The Influence of Consolidation Confining Pressure

When the cement content is 3.5 g/100 mL, the dynamic shear modulus ratio and the damping ratio of RCS soil under consolidation confining pressure are shown in Figure 25. It can be seen from the diagram that the greater the confining pressure, the higher the modulus ratio attenuation curve; that is, the slower the dynamic shear modulus attenuation. When the rubber content *X* ≤ 20%, different confining pressures have little effect on the attenuation law of the modulus ratio. This is because the low rubber content leads to the strong curing effect of cement, which leads to the low deformation ability of the rubber and dislocation between the RCS soil particles. When the rubber content *X* > 20%, the confining pressure has a significant effect on the attenuation of the modulus ratio. This is because the increase in confining pressure makes the deformation of rubber greater, so that the structure of the sample becomes more compact and the particles are not as easy to move so that the modulus decays slowly.

The effect of rubber content on the damping ratio of the RCS soil is shown in Figure 25. When the rubber content *X* ≤ 20%, the confining pressure has little effect on the damping ratio, and the damping ratio curve almost coincides. When the rubber content *X* > 20%, the influence of confining pressure on the damping ratio gradually becomes obvious. It can be seen from the curve in the diagram that the greater the confining pressure, the smaller the damping ratio. This is because when the rubber content is low, the stiffness of the sample is larger, so that the confining pressure has little effect on the sample, and the damping ratio is almost unchanged. When the rubber content is high, because the deformation ability of the sample is strong, the increase in confining pressure makes the deformation of rubber particles greater, so that the damping ratio of the sample with greater confining pressure becomes smaller.

#### 4.3.3. Influence of Cement Content

Figure 26 shows the effect of variations in cement content on the dynamic shear modulus ratio and the damping ratio of RCS soil, along with shear strain amplitude, when the rubber content is 20%. It can be seen from the figure that with the increase in cement content, the attenuation curve of the modulus ratio is higher; that is, the attenuation of the modulus is slower, which is due to the solidification of the cement, creating cementation between particles and an increase in shear resistance. For changes in the damping ratio, the increase in cement content will reduce the damping ratio. This is because the cementation and solidification of cement will increase the contact points between the particles; the particles are not easy to move, resulting in an increase in the stiffness of the RCS soil, a decrease in deformation capacity, a decrease in energy consumption, and a decrease in the damping ratio.

#### 4.3.4. Fitting the Curve Parameters of the Modulus Ratio and Damping Ratio

In this paper, the Martin–Davidenkov model [33] with three parameters is selected to describe the dynamic shear modulus of the RCS soil, and the empirical formula proposed by Chen Guoxing [48] is used to fit and analyze the variation of damping ratio of RCS soil with shear strain amplitude. Figure 27 shows the variation of the fitting parameters A, B, and *γ*_0_, with rubber content in the Martin–Davidenkov model. The values of parameters A and B fluctuate within a certain range with the change in rubber content under different consolidation confining pressures. Therefore, the fitting parameters, A and B, of the RCS soil with a rubber content from 10% to 50% can be taken as 1.1 and 0.4. The reference shear strain, *γ*_0_, basically increases with the increase in rubber content, mainly because the increase in rubber content in the RCS soil makes the attenuation of the dynamic shear modulus of the sample slow down so that the value of the reference shear strain increases. The results show that the *G*/*G*_max_~*γ*_a_ curve of RCS soil can be fitted using the Martin–Davidenkov model.

The change in the fitting parameters, *λ*_min_, *λ*_0_, and *β* with rubber content in the empirical fitting formula is shown in Figure 28. Three phenomena can be identified from the figure. First, the initial damping ratio, *λ*_min_, increases exponentially with the increase in rubber content; the smaller the confining pressure is, the faster the growth rate of *λ*_min_ is. In addition, with the increase in rubber content, the influence of confining pressure on the initial damping is more obvious. Second, parameter *λ*_0_ characterizes the increased amplitude of the damping ratio in the strain range of 10^−6^ to 10^−1^, which is mainly derived from the results of the resonant column test, so that the fitting results fluctuate greatly, and we cannot obtain a unified law. Third, parameter *β* is the shape parameter that describes the increase in the damping ratio. Similar to *λ*_0_, it is mainly obtained by extrapolation, so that the deviation is large. According to the above analysis, it can be seen that *λ*_0_ and *β* are mainly obtained by extrapolation, and there is a certain error. Therefore, only the fitting formula of the initial damping ratio, *λ*_min_, with rubber content and consolidation confining pressure in this test is given as follows:(13)$$\lambda_{\min}=k\cdot e^{n\cdot X}$$
where *λ*_min_ is the initial damping ratio, *K* and *n* are the fitting parameters, and *X* is the rubber content.

The fitting curve is shown in Figure 29 and the fitting parameters *K* and *n* are shown in Table 5. It can be seen that parameter *n* is a constant of 0.035, and parameter *K* increases linearly with the increase in confining pressure. There is a relationship between the effective confining pressure, *σ*_0_, and the parameter, *K*:(14)$$K=-4\times 10^{-5}\cdot \sigma_{0}+0.014$$
where *K* is the fitting parameter and *σ*_0_ is the effective confining pressure.

It can be seen that the fitting formula of the initial damping ratio, *λ*_min_, with the rubber content and consolidation confining pressure in this test, is:(15)$$\lambda_{\min}=\left(-4\times 10^{-5}\cdot \sigma_{0}+0.014\right)\cdot e^{0.035\cdot X}$$

The comparison between the predicted value and the actual value of the initial damping ratio of the RCS soil under different rubber contents and cement contents is shown in Figure 30. It can be seen from the diagram that the scatter points are evenly distributed at around 45°. The fitting R^2^ reaches 0.988, which indicates that the predicted value is close to the measured value. It can be considered that the empirical formula of the initial damping ratio, *λ*_min_, proposed in this section is reliable.

## 5. Main Conclusions

The failure mode, Mohr’s circle, and the stress–strain relationship of the RCS soil were analyzed via an undrained shear test. Finally, the variation law of the dynamic shear modulus, *G*, and the damping ratio, *λ*, of the RCS soil was explored via a resonant column test, and the attenuation of normalized modulus *G/G*_max_ and the growth law of the damping ratio were analyzed. The following conclusions were obtained through an analysis of the experimental results.

Firstly, the addition of rubber particles in the RCS soil changes its static characteristics. With an increase in rubber content, the deviatoric stress at failure of RCS soil gradually decreases, the stress–strain curve no longer peaks, and the specimen gradually changes from the strain-softening type to the hardening type. As the cement content increases, the cementation between particles is more solid, and the deviatoric stress at the failure of the RCS soil is significantly increased, which greatly improves the shear strength of the RCS soil. The internal friction angle of the RCS soil increases with the increase in rubber content but decreases with the increase in cement content. The cohesion of the RCS soil decreases with the increase in rubber content but increases with the increase in cement content.

Secondly, as the cement–rubber content increases, the initial elastic modulus *E*_0_ of the RCS soil decreases in power function and gradually tends toward being linear; the lower the confining pressure, the smaller the initial deformation modulus of the RCS soil. According to the change rule of the initial elastic modulus *E*_0_ with rubber content when the cement content is 3.5 g/100 mL, the empirical formula of the initial elastic modulus, *E*_0_, of the RCS soil is proposed, and the recommended values for the calculation parameters under different working conditions are given.

Thirdly, the maximum dynamic shear modulus *G*_max_ of the RCS soil decreases with the increase in rubber content and increases with the increase in cement content and consolidation confining pressure, which indicates that adding rubber to sand will reduce its shear strength while adding cement to sand will increase its shear strength. The cement added to the RCS soil compensates for the decrease in soil shear strength by adding rubber. According to the variation law of the maximum dynamic shear modulus, *G*_max_, with rubber content when the cement content is 3.5 g/100 mL, the empirical formula of the RCS soil *G*_max_ is proposed, and the recommended values for the calculation parameters under different working conditions are given.

Finally, as the rubber content increases, the modulus ratio curve of the RCS soil is higher, the modulus attenuation is slower, and the damping ratio curve is higher. The greater the cement content, the slower the modulus attenuation, and the lower the damping ratio curve. The increase in consolidation confining pressure makes the dislocation between particles difficult, therefore the modulus decays slowly, the modulus ratio curve becomes higher, and the damping ratio becomes smaller.

However, according to the different usage conditions, it is necessary to further study the influence of different higher cement contents and curing time on the failure strength, internal friction angle, and cohesion of the mixture, so as to further improve the shear strength of the mixtures while ensuring its deformation resistance.

## Figures and Tables

**Figure 1 materials-15-08938-f001:**
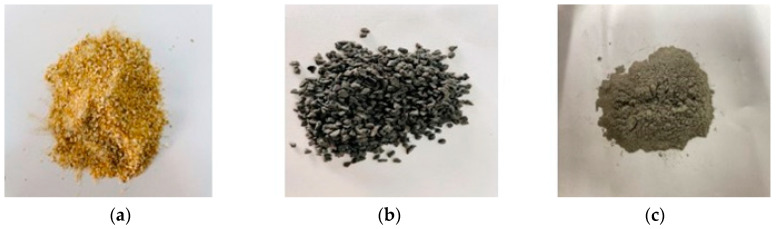
The materials used in the test. (**a**) Fujian standard sand, (**b**) Waste tire rubber particles, (**c**) Nanjing Yuhua cement.

**Figure 2 materials-15-08938-f002:**
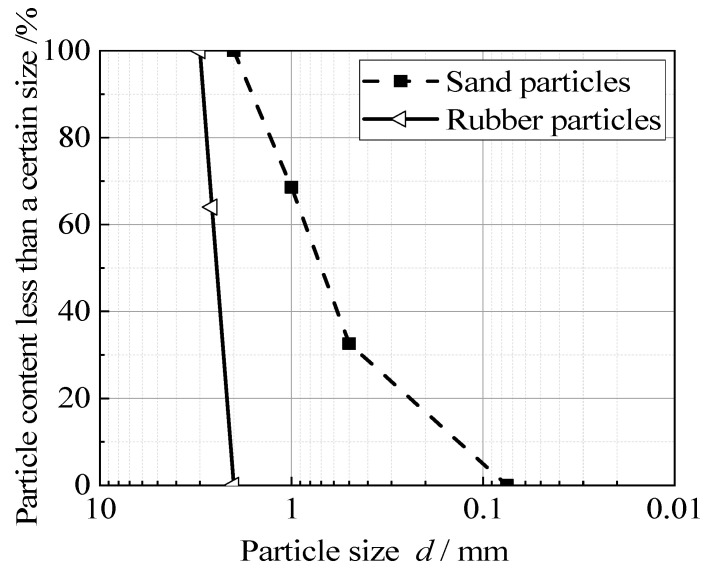
Grading curves.

**Figure 3 materials-15-08938-f003:**
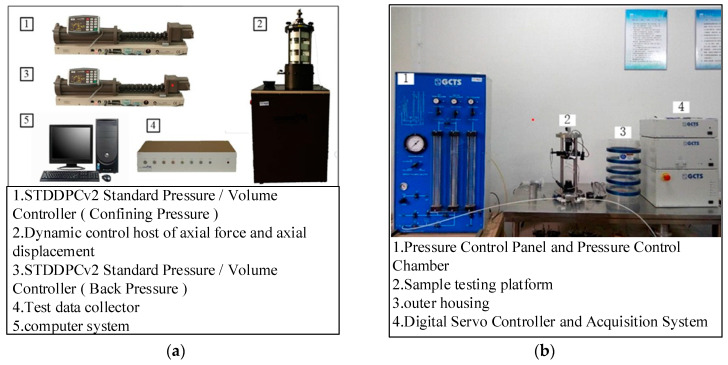
Test instruments used in the current study. (**a**) GDS dynamic triaxial test apparatus, (**b**) TSH-100 type resonant column testing system.

**Figure 4 materials-15-08938-f004:**
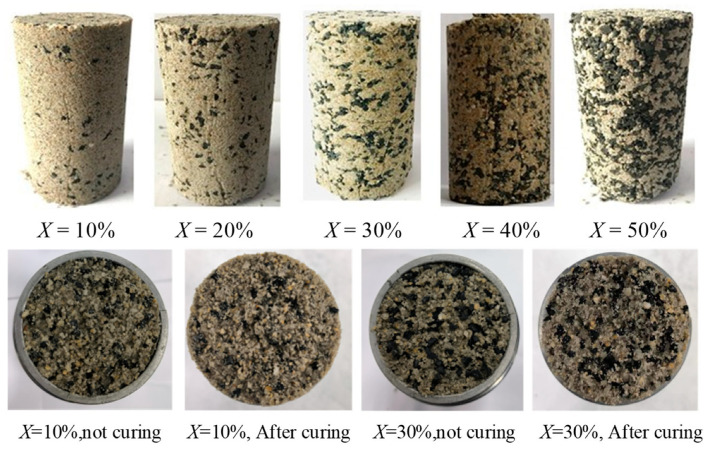
RCS soil samples with rubber contents from 10% to 50%, along with their cross-sections.

**Figure 5 materials-15-08938-f005:**
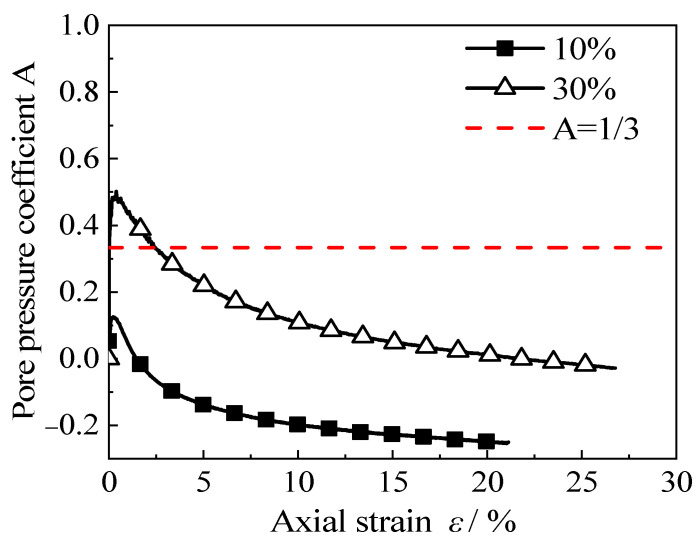
Variation curve of pore pressure coefficient A with the axial strain when *C* = 3.5 g/100 mL.

**Figure 6 materials-15-08938-f006:**
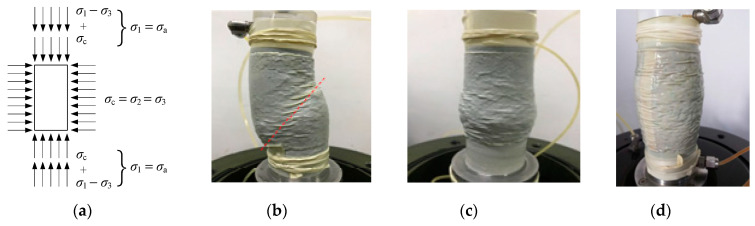
Stress State and Failure Mode of Specimens with a cement content of 3.5 g/100 mL. (**a**) Sample stress state, (**b**) Failure mode with X = 10%, (**c**) Failure mode with X = 20%, (**d**) Failure mode with X = 30%.

**Figure 7 materials-15-08938-f007:**
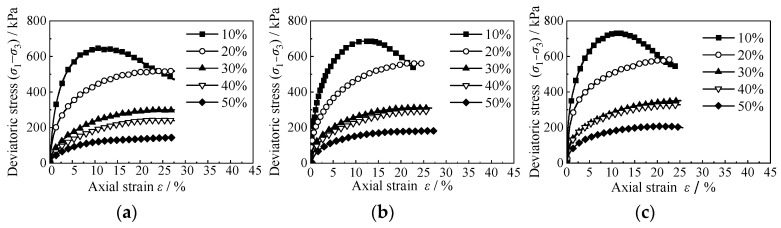
Stress–strain curves of different rubber contents. (**a**) *C* = 3.5 g/100 mL, σ′m = 50 kPa, (**b**) *C* = 3.5 g/100 mL, σ′m = 100 kPa, (**c**) *C* = 3.5 g/100 mL, σ′m = 150 kPa.

**Figure 8 materials-15-08938-f008:**
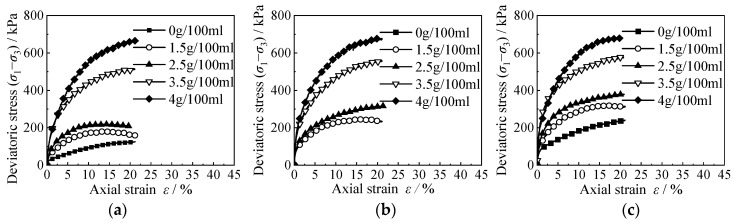
The stress–strain curves when using different cement contents. (**a**) *C* = 3.5 g/100 mL, *σ′*_m_ = 50 kPa, (**b**) *C* = 3.5 g/100 mL, *σ*′_m_ = 100 kPa, (**c**) *C* = 3.5 g/100 mL, *σ′*_m_ = 150 kPa.

**Figure 9 materials-15-08938-f009:**
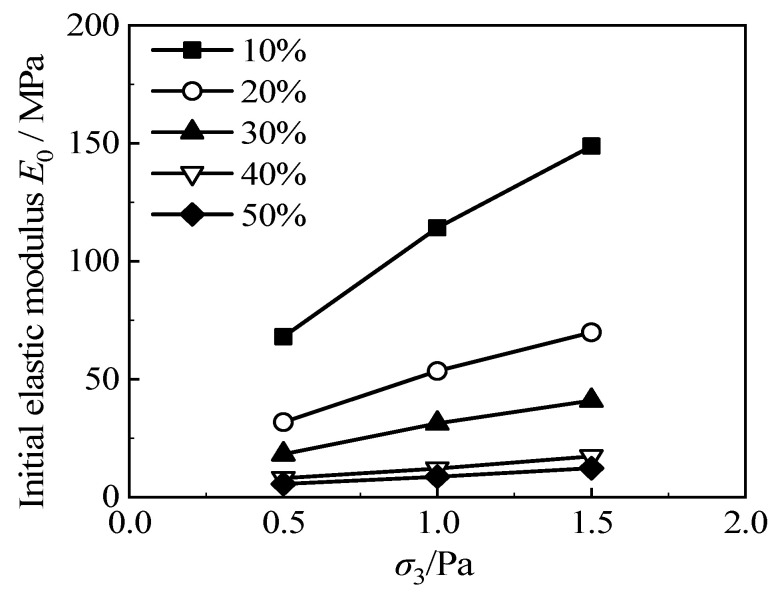
The *E*_0_~*σ*_3_/Pa curve of RCS soil when the cement content is 3.5 g/100 mL.

**Figure 10 materials-15-08938-f010:**
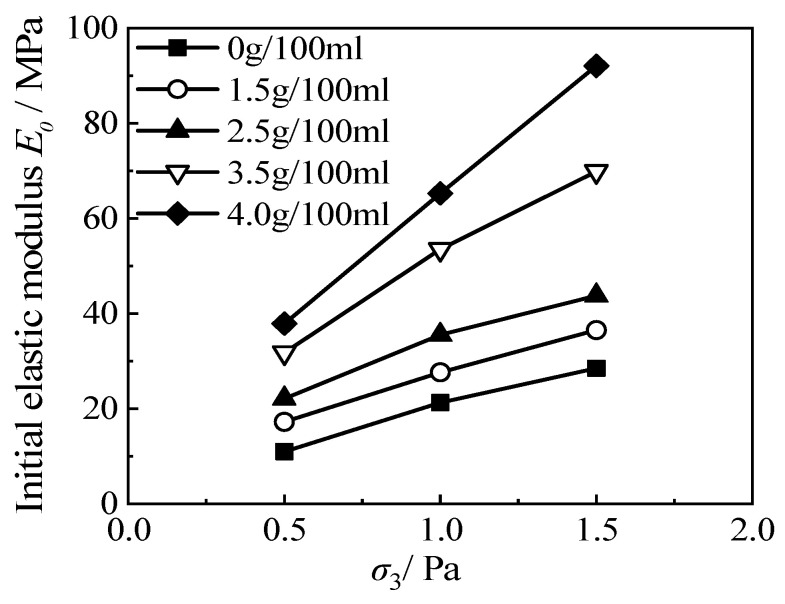
The *E*_0_~*σ*_3_/Pa curve of RCS soil with 20% rubber content.

**Figure 11 materials-15-08938-f011:**
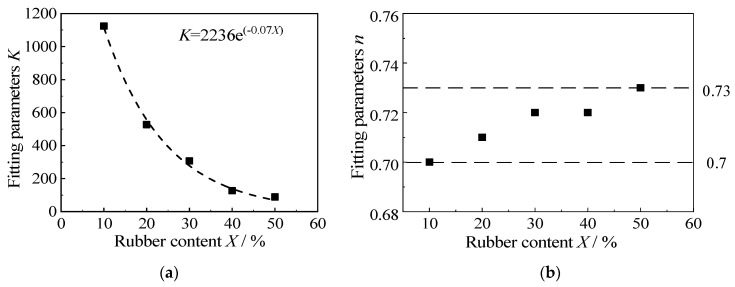
The relationship between the fitting parameters, *K* and *n*, and the cement content. (**a**) Fitting parameters *K*, (**b**) Fitting parameters *n*.

**Figure 12 materials-15-08938-f012:**
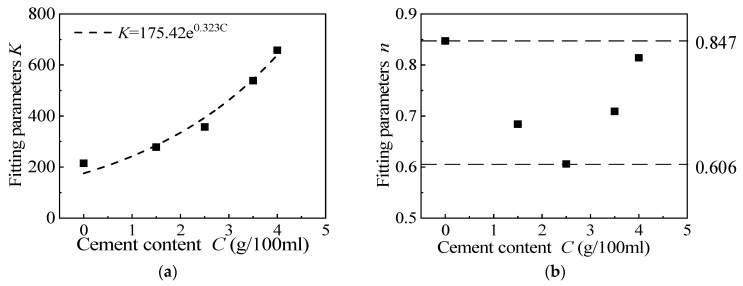
The relationship between the fitting parameters *K* and *n*, according to cement content. (**a**) Fitting parameters *K*, (**b**) Fitting parameters *n*.

**Figure 13 materials-15-08938-f013:**
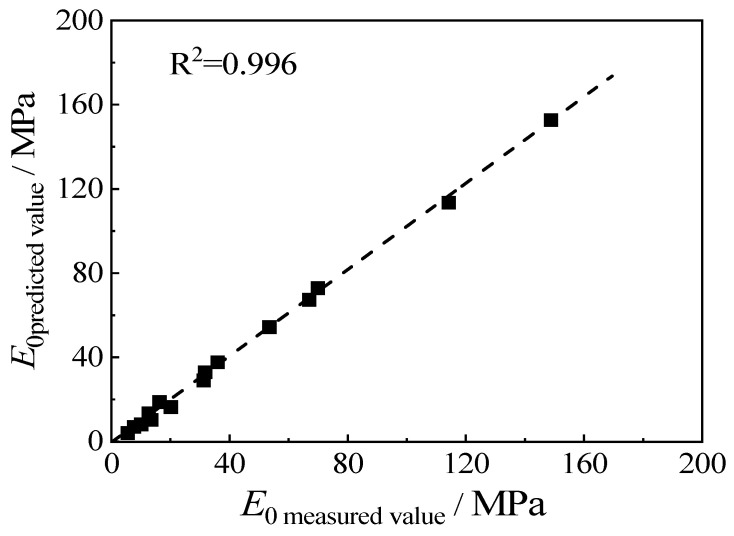
The relationship between the experimental value and the predicted value of the initial elastic modulus.

**Figure 14 materials-15-08938-f014:**
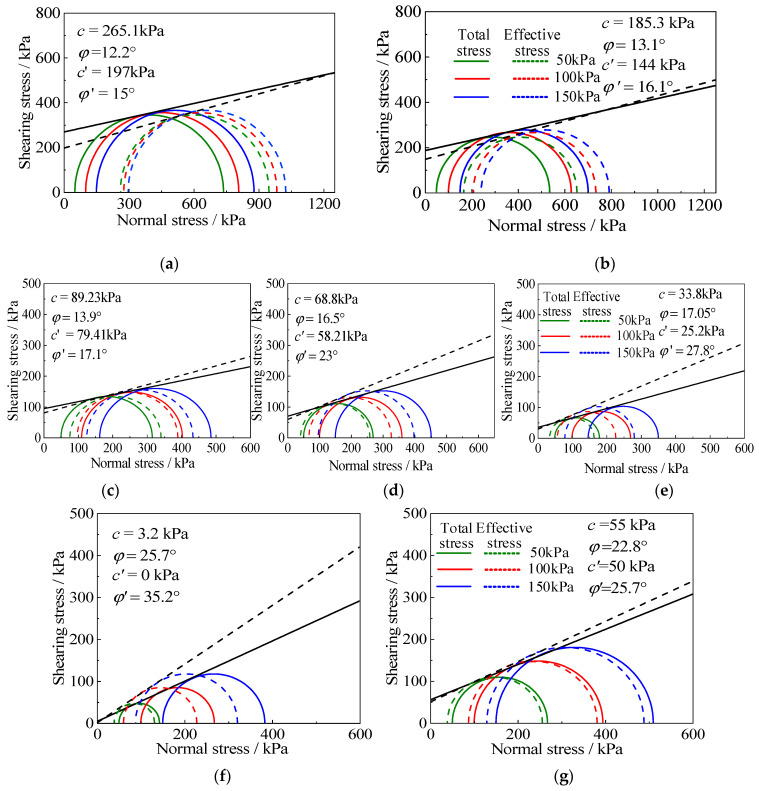
Mohr’s circle and the corresponding shear strength envelope. (**a**) *X* = 10%, *C* = 3.5 g/100 mL, (**b**) *X* = 20%, *C* = 3.5 g/100 mL, (**c**) *X* = 30%, *C* = 3.5 g/100 mL, (**d**) *X* = 40%, *C* = 3.5 g/100 mL, (**e**) *X* = 50%, *C* = 3.5 g/100 mL, (**f**) *X* = 20%, *C* = 0 g/100 mL, (**g**) X = 20%, C = 2.5 g/100 mL.

**Figure 15 materials-15-08938-f015:**
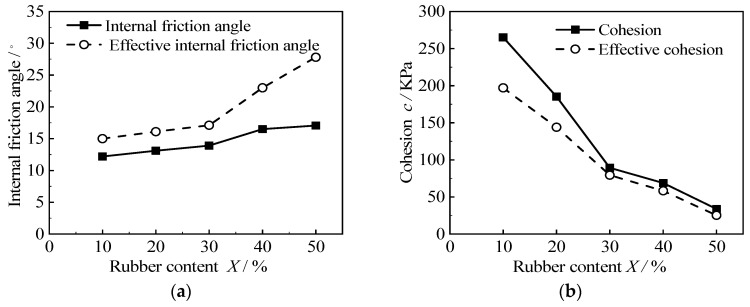
The relationship between the shear strength index of RCS soil and rubber content when *C* = 3.5 g/100 mL. (**a**) Relationship between rubber content and internal friction angle, (**b**) Relationship between Rubber Content and Cohesion.

**Figure 16 materials-15-08938-f016:**
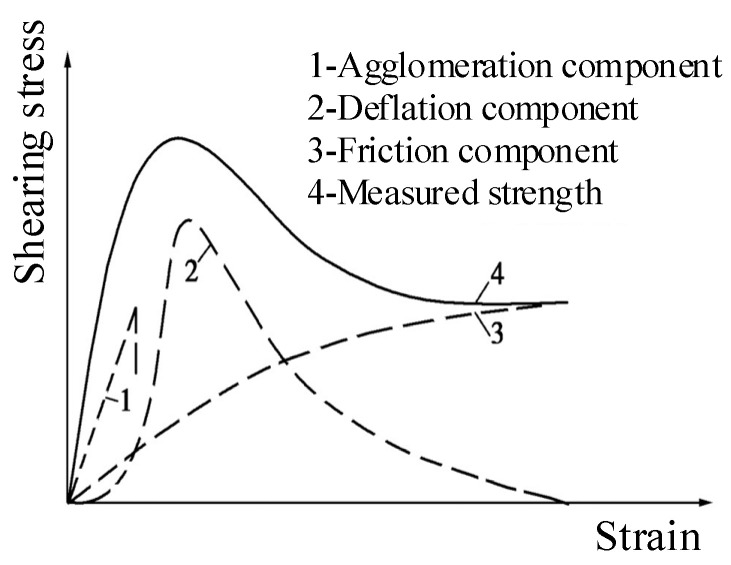
The shear strength mechanism of the soil [39].

**Figure 17 materials-15-08938-f017:**
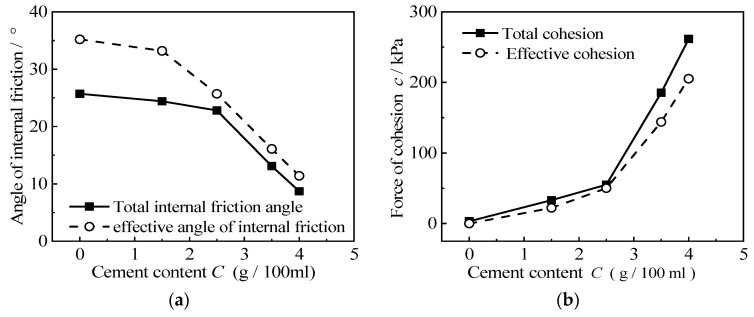
The relationship between the shear strength index of RCS soil and cement content when *X* = 20%. (**a**) Relationship between cement content and internal friction angle, (**b**) Relationship between cement content and cohesion.

**Figure 18 materials-15-08938-f018:**
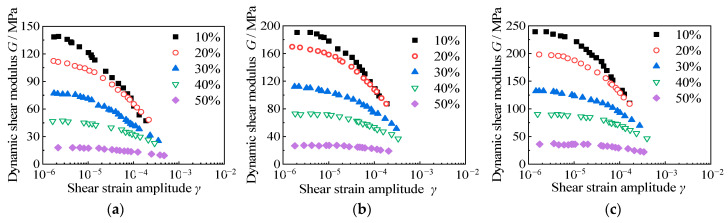
The relationship between the dynamic shear modulus and shear strain amplitude of RCS soil with different rubber contents. (**a**) σ′m = 50 kPa, *C* = 3.5 g/100 mL, (**b**) σ′m = 100 kPa, *C* = 3.5 g/100 mL, (**c**) σ′m = 150 kPa, *C* = 3.5 g/100 mL.

**Figure 19 materials-15-08938-f019:**
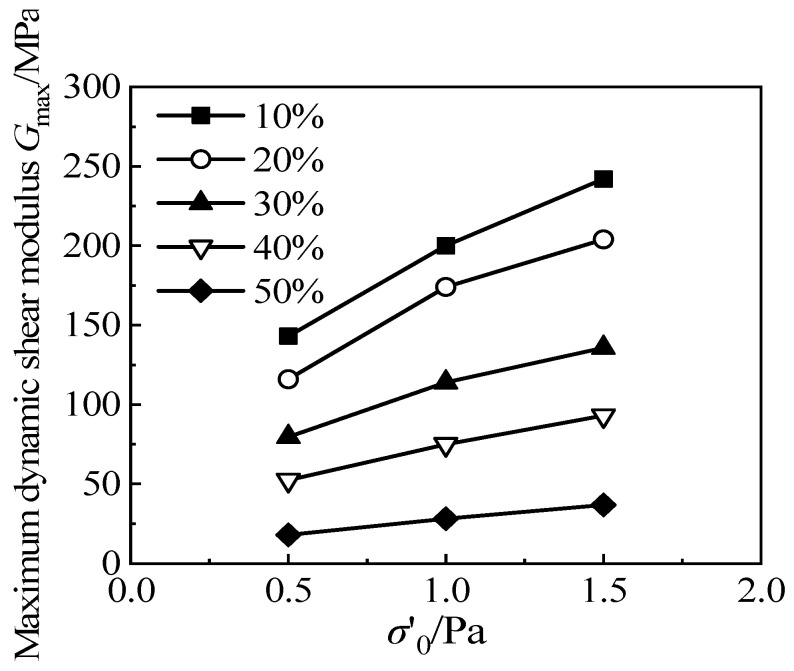
The relationship between *G*_max_ and *σ′*_0_/Pa with different rubber content.

**Figure 20 materials-15-08938-f020:**
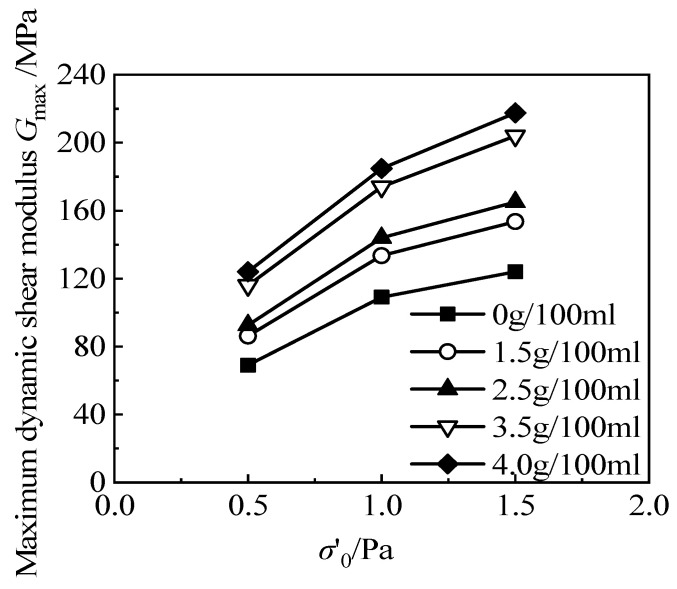
The relationship between *G*_max_ and *σ*′_0_/Pa, with different cement contents.

**Figure 21 materials-15-08938-f021:**
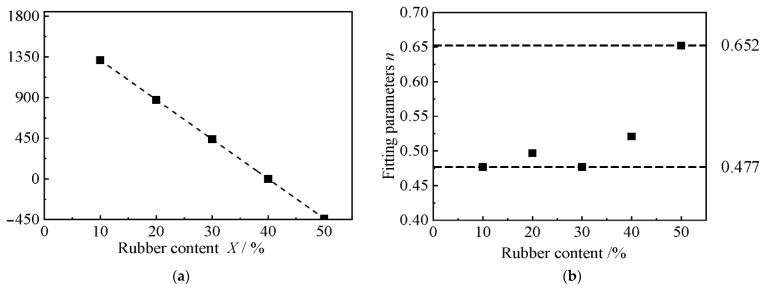
The relationship between fitting parameters *K* and *n* and rubber content. (**a**) Fitting parameters *K*, (**b**) Fitting parameters *n*.

**Figure 22 materials-15-08938-f022:**
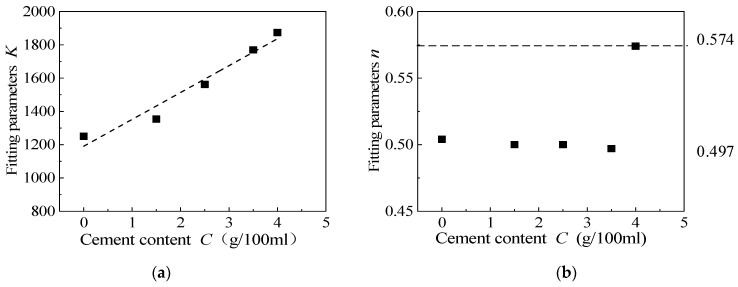
The relationship between fitting parameters *K* and *n* and cement content. (**a**) Fitting parameters *K*, (**b**) Fitting parameters *n*.

**Figure 23 materials-15-08938-f023:**
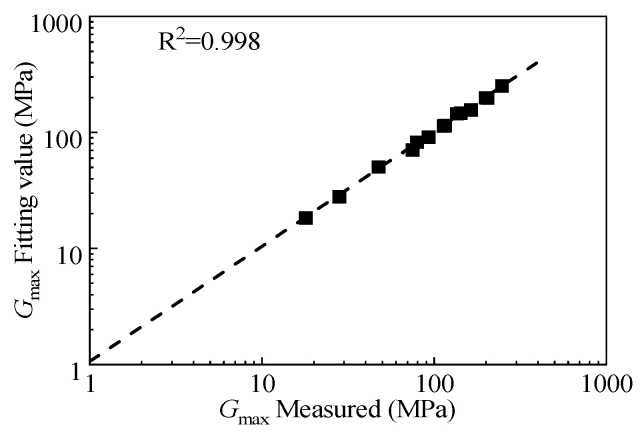
The relationship between the experimental value and fitting value of the maximum dynamic shear modulus.

**Figure 24 materials-15-08938-f024:**
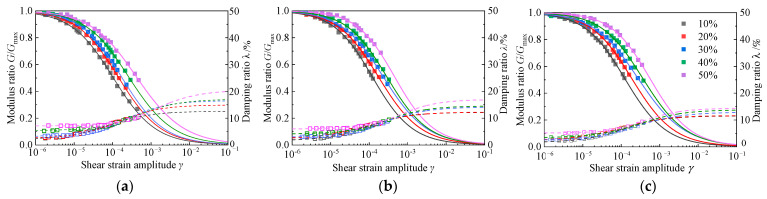
The *G*/*G*_max_~*γ* and *λ*~*γ* curves of RCS soil with different rubber contents. (**a**) *C* = 3.5 g/100 mL, σ′m = 50 kPa, (**b**) *C* = 3.5 g/100 mL, σ′m = 100 kPa, (**c**) *C* = 3.5 g/100 mL, σ′m = 150 kPa.

**Figure 25 materials-15-08938-f025:**
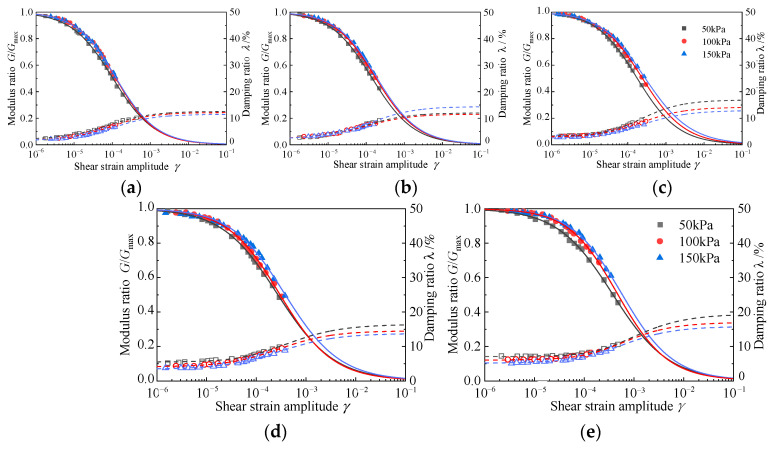
The *G*/*G*_max_~*γ* and *λ*~*γ* curves of the RCS soil under different confining pressures. (**a**) *C* = 3.5 g/100 mL, *X* = 10%, (**b**) *C* = 3.5 g/100 mL, *X* = 20%, (**c**) *C* = 3.5 g/100 mL, *X* = 30%, (**d**) *C* = 3.5 g/100 mL, *X* = 40%, (**e**) *C* = 3.5 g/100 mL, *X* = 50%.

**Figure 26 materials-15-08938-f026:**
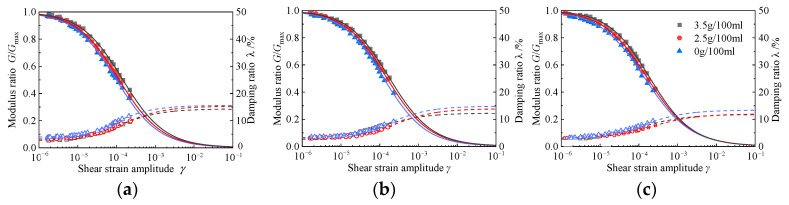
The *G/G*_max_~*γ* and *λ*~*γ* curves of the RCS soil with different cement contents. (**a**) *σ*′_m_ = 50 kPa, *X* = 20%, (**b**), *σ*′_m_ = 100 kPa, *X* = 20%, (**c**) *σ*′_m_ = 150 kPa, *X* = 20%.

**Figure 27 materials-15-08938-f027:**
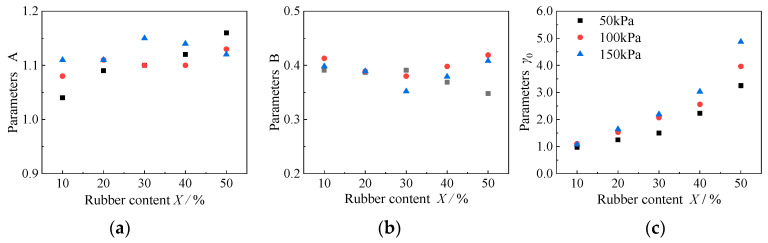
Variation of the model parameters, with the rubber content of the Martin–Davidenkov Model. (**a**) Parameters A, (**b**) Parameters B, (**c**) Parameters *γ*_0_.

**Figure 28 materials-15-08938-f028:**
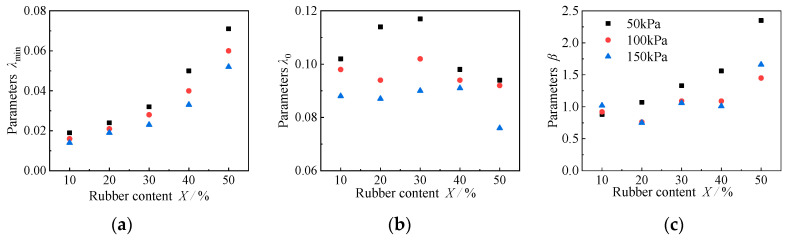
Variations of model parameters with the rubber content and the damping ratio. (**a**) Parameters *λ*_min_, (**b**) Parameters *λ*_0_, (**c**) Parameters *β*.

**Figure 29 materials-15-08938-f029:**
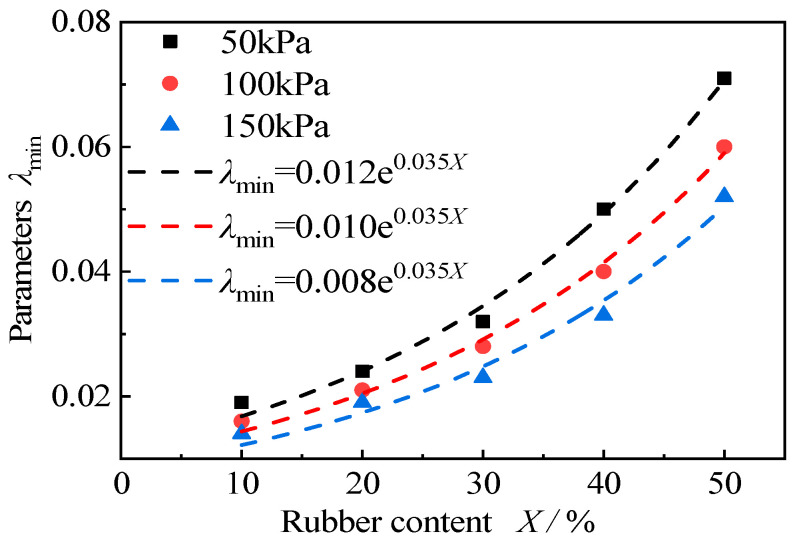
The fitting curve of the initial damping ratio.

**Figure 30 materials-15-08938-f030:**
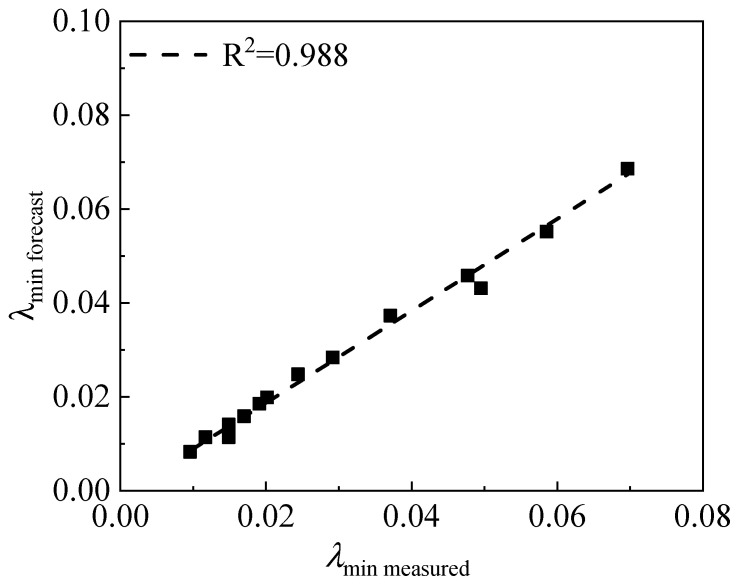
The relationship between the experimental value and the fitting value of the initial damping ratio.

**Table 1 materials-15-08938-t001:** The particle gradation of standard sand.

Name	Value			
Particle size (mm)	2	1	0.5	0.075
Percentage of fine material (%)	100	68.6	32.6	0

**Table 2 materials-15-08938-t002:** Test conditions of the RCS soil.

Rubber Content *X* (%)	Cement Content *C* (g/100 mL)	Relative Compaction *D_r_* (%)	Dry Density *ρ_d_* (g/cm^2^)	Water–Cement Ratio C/W	Curing Age (Day)	Consolidation Confining Pressure *σ*′_m_ (kPa)
0	3.5	70	1.76	1.3	7	50, 100, 150
20	0	70	1.64	1.3	7	50, 100, 150
20	1.5	70	1.64	1.3	7	50, 100, 150
20	2.5	70	1.64	1.3	7	50, 100, 150
20	3.5	70	1.64	1.3	7	50, 100, 150
20	4	70	1.64	1.3	7	50, 100, 150
30	3.5	70	1.53	1.3	7	50, 100, 150
40	3.5	70	1.45	1.3	7	50, 100, 150
50	3.5	70	1.34	1.3	7	50, 100, 150

The water–cement ratio refers to the weight ratio of water to cement in RCS soil.

**Table 3 materials-15-08938-t003:** Deviatoric stress at failure under different working conditions.

Rubber Content	Cement Content	Deviatoric Stress at Failure (*σ*_1_−*σ*_3_)*_f_* (kPa)		
*X* (%)	*C* (g/100 mL)	50 kPa	100 kPa	150 kPa
10	3.5	638.8	681.5	722.3
20	0	112.5	168.0	214.1
20	1.5	176.7	244.0	314.2
20	2.5	215.6	292.4	395.1
20	3.5	484.8	525.3	552.1
20	4.0	623.4	643.5	661.0
30	3.5	273.3	291.2	316.9
40	3.5	218.5	264.5	302.4
50	3.5	129.7	164.9	195.9

**Table 4 materials-15-08938-t004:** Shear strength index of RCS soil.

Rubber Content	Cement Content	Internal Friction Angle	Effective Internal Friction Angle	Cohesion	Effective Cohesion
*X* (%)	*C* (g/100 mL)	*φ* (°)	*φ*′ (°)	*c* (kPa)	*c*′ (kPa)
10	3.5	12.2	15	265.1	197
20	0	25.7	35.2	3.2	0
20	1.5	24.4	33.2	33.1	22.2
20	2.5	22.8	25.7	55	50
20	3.5	13.1	16.1	185.3	144
20	4.0	8.7	11.4	261.7	205.3
30	3.5	13.9	17.1	89.23	79.41
40	3.5	16.5	23	68.8	58.2
50	3.5	17.05	27.8	33.8	25.2

**Table 5 materials-15-08938-t005:** The fitting parameters, *K* and *n*.

Effective Confining Pressure *σ*_0_	50 kPa	100 kPa	150 kPa
*K*	0.012	0.01	0.008
*n*	0.035	0.035	0.035

## Data Availability

Data that support the findings of this study are available from the corresponding author upon reasonable request.

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
