# Peer review of "Experimental Static and Dynamic Characteristics of Recycled Waste Tire Rubber Particle–Cement–Sand Composite Soil"

_materials, 2022, doi:10.3390/ma15248938_

Round 1

Author Response

Question 1: Abstract. There is important information missing in the abstract. what are the variables and their proportions? what is the number of mixes? etc.

Response: Thank the reviewers for their suggestions. According to this comment, the specific content is that the variables studied in this paper are the influence of cement content, rubber content and effective consolidation confining pressure on cement rubber sand mixture. Cement content changes from 0g / 100 ml to 4g / 100 ml. The rubber content is from 10 % to 50 %, and the effective consolidation confining pressure is 50, 100 and 150kpa. A total of 27 working conditions were completed. The above information has been added into abstract. (Lines 15-19)

Question 2: Line 12 – the location of “in this paper” is not correct in the sentence. Please rephrase the first sentence.

Response: Thank the reviewers for their suggestions. The author has made modifications and marks in the abstract part. (Line 14)

Question 3: Introduction: Last paragraph should be expanded further to emphasize the importance of the work.

Response: Thanks to the suggestions of reviewers. According to this suggestion, we have modified and emphasize the importance of the work in the last paragraph of the introduction (Lines 93-97)

Question 4: Line 74. You can’t start a sentence this way “methods. and they pointed out”. Also the punctuation is not correct. Throughout the paper, the punctuation must be corrected.

Response: According to this suggestion, we have revised the sentence. (Line 84)

Question 5: Section 2 test scheme. You should start with the “materials” section, and then move to the “test instruments”.

Response: Thank the reviewers for their suggestions. The author has rewrite this section. (Lines 109-128)

Question 6: Fig 4 is too small. Please include as well in the figure the % cumulative passing in a table form.

Response: According to this comment, the author has made modifications and marks. (Fig.2, Fig.4 and Table.1)

Name

Value

Particle size (mm)

2

1

0.5

0.075

Percentage fine (%)

100

68.6

32.6

0

Question 7: What is the fineness modulus of the sand used?

Response: The fineness modulus of sand used in this paper is 2.462.

Name

Value

Particle size(mm)

4.75

2.36

1.18

0.6

0.3

0.15

Cumulative sieve retained percentage of particles in this size(%)

0

0

23.5

58.6

76.2

87.9

The calculation formula is: Mx = (A2.36+ A1.18+ A0.6+ A0.3+ A0.15-5∙A4.75) / (100-A4.75)

Question 8: Line 118 – the verb is missing in the sentence. “Control relative density Dr = 70 %”.

Response: At the suggestion of experts, the author has made modifications and marks in the abstract part. (Line 134)

Question 9: Line 122 – what is the rationale behind selecting a water cement ration of 1.3? any specifications? Which reference are you following?

Response: Thank the reviewers for their suggestions. The cement slurry commonly used in the project is a typical Newtonian fluid when the water-cement ratio is greater than 1.25. The water cement ratio (W / C) of cement paste is greater than 1.25, the yield stress ( τ0 ) is close to zero with the increase of water cement ratio, and the cement paste almost loses its own internal friction. The related reference has been added into paper.

Reference:

[1]Yang Z Q, Hou K P, Guo T T. Study on the effects of different water-cement ratios on the flow pattern properties of cement grouts[C]//Applied Mechanics and Materials. Trans Tech Publications Ltd, 2011, 71: 1264-1267.

Question 10: Table 1 – second row? The cement content is not correct. Please correct it.

Response: Thanks to the reviewer 's advice, it has been modified and marked in the article. (Table.2)

Question 11: Figure 7, 9, 10, 11, 12, 13, 19 and 20 plots are very small. Please enlarge it.

Response: Thanks to the reviewer 's advice, it has been modified and marked in the article.

Question 12: Line 183 – please delete “the” before table 2.

Response: Thanks to the reviewer 's advice, it has been modified and marked in the article.(Line 210)

Question 13: Are the assumptions of linear elasticity still valid? Please provide updates references

Response: Thanks to the reviewers ' questions, the linear elasticity hypothesis is still valid in the small strain range. This paper also studies the deformation in small strain range. The related reference has been added into paper.

Reference:

[1] Xia P, Shao L, Deng W. Mechanism study of the evolution of quasi-elasticity of granular soil during cyclic loading[J]. Granular Matter, 2021, 23(4): 1-15.

Question 14: General comment: figure and tables should be placed after being mentioned in the text and not before.

Response: Thanks to the reviewer 's advice, we have modified and marked in the article.

Question 15: What are the authors explanations that the angle of internal friction decreases when the cement content goes up?

Response: The increase of cement content makes the stiffness of the sample larger, and the influence of confining pressure on it decreases, so that the radius of the mohr's circle under different confining pressures changes little, and the slope of the envelope curve decreases, resulting in the decrease of the internal friction angle. (Lines 313-316)

Question 16: From where the authors came up with equation 8? Any reference?

Response: Thanks to the reviewer 's question. Equation 8 is based on the model of Equation 7. Due to the following test studies, some scholar simplified the Eq. 7 to Eq. 8 to neglect the effect of the relative density. It has been explained in Line 377.

Question 17: Fig 29. If the rubber content goes up well beyond the 50%, what would be the value of lmin in this case? Equation 16 should have an upper limit?

Response: Thanks to the reviewer 's question. In our studies, the rubber content is far more than 50 %. For example, when the maximum rubber content is 100 %, the minimum value of λmin is 0.331 under the confining pressure of 100 kPa. At present, most of the rubber content used in the literature we have reviewed does not exceed 50 % in civil engineering. According to this situation, we only analyzed the test results when the maximum rubber content is less than 50 %.

Question 18: Based on the authors study, what is the optimum % of waste tire rubber that can be used in soil?

Response: Thanks to the reviewer 's question. Due to the different usage condition of the mixtures, the different optimum of waste tire rubber should be defined. In this study, the usage condition has not been considered, and so the optimum of waste tire rubber has not been analyzed in this paper. According to this comment, we will analysis this problem in the following study.

Question 19: I suggest to the authors to add a few sentences about recommendations for future work

Response: Thank you for this suggestion, we have added the suggested works in the last paragraph of this paper. (Lines 599-602)

Reviewer 2 Report

The topic treated by the Authors is certainly worthy of investigation. However, the paper needs several improvements in order to be published.

Lines 84-85

“In order to solve the above problems, this paper proposes to add cement as a cementing curing agent to the RS soil to improve the shear strength of the r RS soil.”

1.     Is the “r” in “the r RS soil” a typo? Please check.

2.     Given the quantity of cement used, in doing this the Authors approach experiments already conducted in a different field, namely that of civil engineering. Geotechnics and civil engineering often study similar problems but, unfortunately, the exchange of information between these two sectors is scarce. Obviously, the aim of a civil engineer is different, as he aims to improve the mechanical characteristics of concrete solids (in this case, by adding rubber particles deriving from waste tires). However, the “ingredients” of the mixtures studied are similar, except for the fact that the Authors do not use coarse aggregates. Therefore, to better frame the topic of the article in the international scientific panorama, it is appropriate that the Authors also mention some works developed in the field of civil engineering. Just to give some examples (but feel free to expand the list): to the knowledge of the reviewer, one of the first authors to have dealt with rubberized concrete is Topçu (Topçu, 1995), while one of the first attempts to exploit the greater damping provided by rubber aggregates for structural purposes is reported in Ferretti and Bignozzi (Ferretti and Bignozzi, 2012).

Topçu, İ.B. The properties of rubberized concretes. Cement Concrete Res., 1995, 25(2): 304–310.

Ferretti, E., Bignozzi, M. Stress and strain profiles along the cross-section of waste tire rubberized concrete plates for airport pavements. Computers, Materials & Continua, 2012, 27(3): 231–274.

Lines 136-138

“In the undrained shear test, the RCS soil with different rubber content shows different mechanical properties and failure modes.

Fig. 6 shows two failure modes of the RCS soil sample under different working conditions.”

Although the title of Section 3 is “Analysis of Undrained Shear Test Results” and the fact that the analysis is actually limited to the undrained shear is remarked in lines 136-137, Figure 6 could be confused with a uniaxial compression test. In fact, a brittle specimen subjected to uniaxial compression test can fail (and usually fails) precisely with the formation of one or more sliding planes. Please provide more information on Figure 6 or use a more appropriate figure, which leaves no doubt about how the load is transmitted to the specimen. Otherwise, the reader may doubt that the authors confuse the fact that the failure mode occurs along a sliding plane (therefore due to the shear strength limits reached) with the fact that the test is a shear test.

Lines 143-144

“When the rubber content X ≥ 20%, the rubber particles will produce large compression deformation under the action of shear”

This should be explained by the use of a figure.

Line 154

“The stress-strain curve of undrained shear test of the RCS soil with different rubber content is drawn in Fig. 7.”

Typically, “stress-strain curve” is used to indicate an axial stress-strain curve, in a uniaxial compression test. Make it clear that these are shear stresses. Also explain how you acquired the strains and how you identified the deviatoric stresses. Please do not say that the testing machine provide them directly but write their formulas.

Lines 182-184

“Three conclusions can be seen from the Table. 2. The first is that with the increase of rubber content, the smaller the stiffness of the sample, the smaller the deviatoric stress at failure.”

A “Deviatoric stress at failure” label should appear in the table.

Lines 235-237

“According to the failure deviatoric stress of RCS soil under different rubber content and cement content, the total molar stress circle and the effective molar stress circle under the corresponding rubber content are drawn on the τ – σ coordinate diagram according to the specification”

Molar stress circle? Is it the “circle of Mohr”, as it appears from Fig. 14? Please check in all the text.

Provide information on identifying total circles and effective circles.

Equation 10

Since the Authors propose this equation as an improvement of Eq. (7), it should give the same expression as Eq. (7) in the absence of rubber (X=0). This means that the coefficient 2480 in Eq. (9) should be a function of the relative compactness of sand, such as K_max, not a coefficient.

Equation 11

Same comment made for Eq. (10).

There should be only one expression for the parameter K, which gives 21.7 K_max for X=0 and C=0.

Author Response

Question 1: Is the “r” in “the r RS soil” a typo? Please check.

Response: Thanks to the questions, we have modified and marked them in the article. (Line 99)

Question 2: Given the quantity of cement used, in doing this the Authors approach experiments already conducted in a different field, namely that of civil engineering. Geotechnics and civil engineering often study similar problems but, unfortunately, the exchange of information between these two sectors is scarce. Obviously, the aim of a civil engineer is different, as he aims to improve the mechanical characteristics of concrete solids (in this case, by adding rubber particles deriving from waste tires). However, the “ingredients” of the mixtures studied are similar, except for the fact that the Authors do not use coarse aggregates. Therefore, to better frame the topic of the article in the international scientific panorama, it is appropriate that the Authors also mention some works developed in the field of civil engineering. Just to give some examples (but feel free to expand the list): to the knowledge of the reviewer, one of the first authors to have dealt with rubberized concrete is Topçu (Topçu, 1995), while one of the first attempts to exploit the greater damping provided by rubber aggregates for structural purposes is reported in Ferretti and Bignozzi (Ferretti and Bignozzi, 2012).

Topçu, İ.B. The properties of rubberized concretes. Cement Concrete Res., 1995, 25(2): 304–310.

Ferretti, E., Bignozzi, M. Stress and strain profiles along the cross-section of waste tire rubberized concrete plates for airport pavements. Computers, Materials & Continua, 2012, 27(3): 231–274.

Response: Thank the reviewers for their suggestions. As a kind of solid waste material, rubber is applied in various engineering fields, and it is also more widely used in civil engineering. The strong compression performance of waste tire rubber particles improves the ductility of the material. The author has modified and marked it in the text (Lines 39-41). The related references are also added into this paper.

Reference:

[1] Ferretti, E., Bignozzi, M. C. Stress and strain profiles along the cross-section of waste tire rubberized concrete plates for airport pavements[J]. Computers Materials and Continua, 2012, 27(3): 231.

[2] Liu, H. B, Wang X. Q., Jiao Y. B., Sha, T. Experimental investigation of the mechanical and durability properties of crumb rubber concrete[J]. Materials, 2016, 9(3): 172.

[3] Topcu, I. B. The properties of rubberized concretes[J]. Cement and concrete research, 1995, 25(2): 304-310.

Lines 136-138

“In the undrained shear test, the RCS soil with different rubber content shows different mechanical properties and failure modes.

Fig. 6 shows two failure modes of the RCS soil sample under different working conditions.”

Question 2: Although the title of Section 3 is “Analysis of Undrained Shear Test Results” and the fact that the analysis is actually limited to the undrained shear is remarked in lines 136-137, Figure 6 could be confused with a uniaxial compression test. In fact, a brittle specimen subjected to uniaxial compression test can fail (and usually fails) precisely with the formation of one or more sliding planes. Please provide more information on Figure 6 or use a more appropriate figure, which leaves no doubt about how the load is transmitted to the specimen. Otherwise, the reader may doubt that the authors confuse the fact that the failure mode occurs along a sliding plane (therefore due to the shear strength limits reached) with the fact that the test is a shear test.

Response: Thanks to the reviewer 's advice, we have modified and marked in the article. (Fig 6)

Fig. (6a) shows the stress of the specimen in the consolidated undrained shear test, from which we can see the load transfer path in the triaxial test.

Lines 143-144

“When the rubber content X ≥ 20%, the rubber particles will produce large compression deformation under the action of shear”

Question 3: This should be explained by the use of a figure.

Response: Thanks to the reviewer 's advice, the author of this sentence ill-considered, we have revised and marked in the text. (Lines 171-174)

Line 154

“The stress-strain curve of undrained shear test of the RCS soil with different rubber content is drawn in Fig. 7.”

Question 4: Typically, “stress-strain curve” is used to indicate an axial stress-strain curve, in a uniaxial compression test. Make it clear that these are shear stresses. Also explain how you acquired the strains and how you identified the deviatoric stresses. Please do not say that the testing machine provide them directly but write their formulas.

Response: Thanks to the reviewer 's advice. The sentence has been revised in Line 182.

Lines 182-184

“Three conclusions can be seen from the Table. 2. The first is that with the increase of rubber content, the smaller the stiffness of the sample, the smaller the deviatoric stress at failure.”

Question 5: A “Deviatoric stress at failure” label should appear in the table.

Response: Thanks to the suggestions of reviewers, we have modified and marked in Table 3.

Lines 235-237

“According to the failure deviatoric stress of RCS soil under different rubber content and cement content, the total molar stress circle and the effective molar stress circle under the corresponding rubber content are drawn on the τ – σ coordinate diagram according to the specification”

Question 6: Molar stress circle? Is it the “circle of Mohr”, as it appears from Fig. 14? Please check in all the text.

Provide information on identifying total circles and effective circles.

Response: Thanks to the mistakes pointed out by the reviewers, we have modified and marked them in the article. The method of identifying the total circle and the effective circle is to draw the total molar stress circle and the effective molar stress circle under the corresponding rubber content on the τ-σ coordinate diagram according to the Standard for Soil Test Methods (GB / T 50123-2019). The related reference had been added into the paper.

Reference:

[1] GB / T 50123-2019, soil test method standard [S]. GB / T 50123-2019, soil test method standard [S].

[2] Tatsuoka, F., Teachavorasinskun, S., Kong, X. J., Abe, F., KIM, Y., & PARK, C. Elastic deformation properties of geomaterials[J]. Soils and Foundations, 1992, 32(3): 26-46.

[3] Qian J.H., & Yin Z. Z, Geotechnical principle and calculation [M]. China Water Conservancy and Hydropower Press, 1996.

Equation 10

Question 7: Since the Authors propose this equation as an improvement of Eq. (7), it should give the same expression as Eq. (7) in the absence of rubber (X=0). This means that the coefficient 2480 in Eq. (9) should be a function of the relative compactness of sand, such as Kmax, not a coefficient.

Response: Thanks to the reviewer 's advice. Due to the following test studies, some scholar simplified the Eq. (7) to Eq. (8) to neglect the effect of the relative density. It has been explained in Line 377. In this study, Eq. (9) is corresponding to Eq. (8).

Equation 11

Same comment made for Eq. (10).

Question 8: There should be only one expression for the parameter K, which gives 21.7 Kmax for X=0 and C=0.

Response: Thanks to the reviewer 's advice. Formula (11) and formula (9) was revised to be one expression for the parameter K, is as follows:

K =1752 -43.8∙X + 208∙C

Reviewer 3 Report

The paper is well organized and has good information that readers may find very important. The following recommendations should be implemented before publication to improve the quality of the manuscript.

  • The authors were invited to add an image of a different percentage failure in figure 6 to show that 20% changed the failure mode.
  • To obtain Fig. 4, provide more information (standards) about the tests that were performed.
  • Explain how the authors can justify the uniformity of rubber particles among sand particles and the 7-day curing time. (Some cross-section photos can be added to Fig. 5)
  • Double check the paper format; figures 1 and 2 can be combined using (a) and (b), ovoid to spare a figure between two pages, as in the case of fig. 7.
  • In Section (4.3.1) more analysis should be provided about the influence of rubber percentage on compression behavior, absorbed energy, failure mechanisms, the rubber-cement interface, etc., and correlate results with previous studies (https://doi.org/10.1016/j.conbuildmat.2022.128820 ; https://doi.org/10.1016/j.jclepro.2020.123074 ; https://doi.org/10.1016/j.conbuildmat.2021.123105.
  • It is strongly suggested that the conclusion be rewritten to be more direct about the obtained results. 

Author Response

Question 1: The authors were invited to add an image of a different percentage failure in figure 6 to show that 20% changed the failure mode. To obtain Fig. 4, provide more information (standards) about the tests that were performed.

Response: Thanks to the reviewer 's advice, we have modified and marked in Fig 6.

For the information in Figure 4, the first reviewer raised the same question, I replied this question above and will not repeat it here.

Question 2: Explain how the authors can justify the uniformity of rubber particles among sand particles and the 7-day curing time. (Some cross-section photos can be added to Fig. 5)

Response: Thanks to the reviewer 's questions, we have added the cross-section photos in Fig 4. From the cross-sectional view of the sample in Fig.4, it can be seen that the rubber is relatively uniform between the sand particles. The cross-sectional view of the cured sample obviously looks more condensed than the uncured sample.

Fig.  4 RCS soil sample with rubber content from 10% to 50% and cross section

Question 3: Double check the paper format; figures 1 and 2 can be combined using (a) and (b), ovoid to spare a figure between two pages, as in the case of fig. 7.

Response: Thanks to the advice of the reviewer, we have revised the related figures according to this comment.

Question 4: In Section (4.3.1) more analysis should be provided about the influence of rubber percentage on compression behavior, absorbed energy, failure mechanisms, the rubber-cement interface, etc., and correlate results with previous studies (https://doi.org/10.1016/j.conbuildmat.2022.128820; https://doi.org/10.1016/j.jclepro.2020.123074; https://doi.org/10.1016/j.conbuildmat.2021.123105.)

Response: Thanks to the reviewer ' s advice. The related references have been added in this paper, and some more analysis has been added in Line 458-478.

Reference:

[1] Abbassi F, Ahmad F. Behavior analysis of concrete with recycled tire rubber as aggregate using 3D-digital image correlation[J]. Journal of Cleaner Production, 2020, 274: 123074.

[2] Kang J, Liu Y, Yuan J, et al. Effectiveness of surface treatment on rubber particles towards compressive strength of rubber concrete: A numerical study on rubber-cement interface[J]. Construction and Building Materials, 2022, 350: 128820.

[3] Yu Z, Tang R, Li F, et al. Experimental study and failure criterion analysis on combined compression-shear performance of rubber concrete (RC) with different rubber replacement ratio[J]. Construction and Building Materials, 2021, 288: 123105.

Question 5: It is strongly suggested that the conclusion be rewritten to be more direct about the obtained results. 

Response: According to this comment, we have revised the conclusion. (Lines 573-602)

Other minor changes were directly revised and marked in red.

Round 2

Reviewer 1 Report

The authors properly addressed my comments and significantly improved the manuscript quality. I dont have further comments but I urge the authors to proofreading their papers one more time.

Author Response

We have revised our paper one more time. Thanks

Reviewer 2 Report

The Authors have significantly improved the document. However, more fixes are needed.

Pag. 7

“When X ≥ 20 %, with the increase of axial strain, pore pressure coefficient of RCS soil first shear dilatation and then shear contraction.”

According to what previously explained by the Authors (“Pore pressure coefficient A is constantly changing in undrained shear test, for elastic body A = 1/3, for dilatancy soil A < 1/3, for shear shrinkage soil A > 1/3”), it would seem the opposite:

When X ≥ 20 %, with the increase of axial strain, pore pressure coefficient of RCS soil first indicates shear contraction (A > 1/3) and then shear dilatation (A < 1/3).

“The stress state of the specimen during the test is shown in Fig. 6(a).”

Where is the shear stress in Fig. 6(a)? This is the stress state of a specimen during the tri-axial compression test, not during the undrained shear test. As commented in the previous review, the Authors seem to present the results of (tri-axial) compression tests, not those of undrained shear tests. Please do not confuse the fact that the failure mode occurs along a sliding plane (therefore due to the shear strength limits reached) with the fact that the test is a shear test. Specimens subjected to both uniaxial and tri-axial compression tests fail with the formation of sliding planes although no shear stress is applied to the specimen heads. As it appears that the tests performed by the Authors were tri-axial compression tests, the Authors should change the title of Section 3 and the first line of Section 3.1.

Response to Question 4

“Thanks to the reviewer’s advice. The sentence has been revised in Line 182.”

The reviewer actually found no revisions in the new text. This may have happened due to the fact that the tests are actually compression (and not shear) tests. Thus, this indicates once again that the title of Section 3 and the first line of Section 3.1 should be changed. Furthermore, the Authors did not explain how they acquired the strains. This should be added to the text.

Response to Question 4

“mohr” should be capitalized, as it is the surname of a researcher. Please check all text.

Author Response

Pag. 7

Question: “When X ≥ 20 %, with the increase of axial strain, pore pressure coefficient of RCS soil first shear dilatation and then shear contraction.”

According to what previously explained by the Authors (“Pore pressure coefficient A is constantly changing in undrained shear test, for elastic body A = 1/3, for dilatancy soil A < 1/3, for shear shrinkage soil A > 1/3”), it would seem the opposite:

When X ≥ 20 %, with the increase of axial strain, pore pressure coefficient of RCS soil first indicates shear contraction (A > 1/3) and then shear dilatation (A < 1/3).

 Response: Thank you to the reviewer for the errors noted, the reader has made changes in the text lines 160-162 and marked.

Question: “The stress state of the specimen during the test is shown in Fig. 6(a).”

Where is the shear stress in Fig. 6(a)? This is the stress state of a specimen during the tri-axial compression test, not during the undrained shear test. As commented in the previous review, the Authors seem to present the results of (tri-axial) compression tests, not those of undrained shear tests. Please do not confuse the fact that the failure mode occurs along a sliding plane (therefore due to the shear strength limits reached) with the fact that the test is a shear test. Specimens subjected to both uniaxial and tri-axial compression tests fail with the formation of sliding planes although no shear stress is applied to the specimen heads. As it appears that the tests performed by the Authors were tri-axial compression tests, the Authors should change the title of Section 3 and the first line of Section 3.1.

 Response: Thanks to the reviewer 's suggestion, the reader has changed the title of Section 3 and the first line of Section 3.1.(Lines150-152)

Response to Question 4

“Thanks to the reviewer’s advice. The sentence has been revised in Line 182.”

Question: The reviewer actually found no revisions in the new text. This may have happened due to the fact that the tests are actually compression (and not shear) tests. Thus, this indicates once again that the title of Section 3 and the first line of Section 3.1 should be changed. Furthermore, the Authors did not explain how they acquired the strains. This should be added to the text.

 Response: Thanks to the reviewer 's advice, the author has modified and marked the title of Section 3 and the first line of Section 3.1, and added the definition of strain. (Lines150-152, lines 121-122.)

Response to Question 4

Question: “mohr” should be capitalized, as it is the surname of a researcher. Please check all text.

 Response: Thank you for the reviewer 's suggestion. Authors have made changes and annotations.

Reviewer 3 Report

Authors should double-check the style and format in several sections of the manuscript. For example:

- in the abstract, ": In order to improve..... In order to explore the influence..."

- In Table 1: Nanjing Tech University, as shown in Error! Reference source not found.. During the test..

- Table 2 style should be revised.

Author Response

Authors should double-check the style and format in several sections of the manuscript. For example:

Question: - in the abstract, ": In order to improve..... In order to explore the influence..."

Response: Thanks to the reviewer 's advice, We have made changes to the text and marked. (Lines13-16, Line575, Line580, Line593.)

Question: - In Table 1: Nanjing Tech University, as shown in Error! Reference source not found..

During the test.

Response: Thank you to the reviewer for the error noted, We have made changes in the text line 121 and marked.

Question: - Table 2 style should be revised.

Response: Thanks to the reviewer 's advice, We have made changes to Table 2. (Line146-147)  

Other minor changes are directly represented by yellow highlights.